# IMDPrompter: Adapting SAM to Image Manipulation Detection by Cross-View Automated Prompt Learning

**Quan Zhang**[1][*] **Yuxin Qi**[2][*] **Xi Tang**[1]**, Jinwei Fang**[3]**, Xi Lin**[2]**, Ke Zhang**[1][†]**, Chun Yuan**[1][†]
[1]Tsinghua University  [2]Shanghai Jiao Tong University
[3]University of Science and Technology of China
{zhangqua22, ke-zhang19}@mails.tsinghua.edu.cn,
qiyuxin98@sjtu.edu.cn, yuanc@sz.tsinghua.edu.cn

## Abstract

Using extensive training data from SA-1B, the Segment Anything Model (SAM) has demonstrated exceptional generalization and zero-shot capabilities, attracting widespread attention in areas such as medical image segmentation and remote sensing image segmentation. However, its performance in the field of image manipulation detection remains largely unexplored and unconfirmed. There are two main challenges in applying SAM to image manipulation detection: a) reliance on manual prompts, and b) the difficulty of single-view information in supporting cross-dataset generalization. To address these challenges, we develops a cross-view prompt learning paradigm called IMDPrompter based on SAM. Benefiting from the design of automated prompts, IMDPrompter no longer relies on manual guidance, enabling automated detection and localization. Additionally, we propose components such as Cross-view Feature Perception, Optimal Prompt Selection, and Cross-View Prompt Consistency, which facilitate cross-view perceptual learning and guide SAM to generate accurate masks. Extensive experimental results from five datasets (CASIA, Columbia, Coverage, IMD2020, and NIST16) validate the effectiveness of our proposed method.

## 1 Introduction

With the continuous emergence of powerful editing tools Zhang et al. (2025a), image manipulation has become unprecedentedly simple. These new opportunities have sparked the creativity of both benevolent and malicious users. In the past, organizing multimedia misinformation activities required complex skills, with attackers limited to splicing, copying, or deleting objects.Recently, deep learning has advanced at a rapid pace , leading to the development of more user-friendly and powerful image editing tools. As a result, users can now swiftly generate images of fictional characters or create highly convincing deepfakes. Generative models can produce realistic image edits based on natural language prompts, perfectly matching inserted elements with the style and lighting of the environment Avrahami et al. (2023); Nichol et al. (2021).

However, the risks of these tools falling into the wrong hands are evident. In recent years, governments and funding agencies have shown increasing interest in developing forensic tools capable of addressing such attacks, particularly focusing on local image tampering that alters the semantic content of images Le et al. (2021). In response to these challenges, the fields of multimedia forensics and related sciences have rapidly expanded, proposing various methods and tools for image manipulation detection (IMD) and localization Guillaro et al. (2023). Despite significant progress in this area, the performance of state-of-the-art detectors in practical applications remains insufficient, primarily limited by several shortcomings that require further research: a) limited generalization ability; b) limited robustness; c) poor detection performance.

---

[*]These authors contributed equally.
[†]Corresponding authors.

To address the shortcomings of existing IMD methods, we turn our attention to foundational models Kirillov et al. (2023); Radford et al. (2021); Jia et al. (2021); Sharif Razavian et al. (2014). Thanks to large-scale pretraining, foundational models like GPT-4 Achiam et al. (2023), Flamingo Alayrac et al. (2022), and SAM Kirillov et al. (2023) have made significant strides and contributed importantly to societal advancements. Among them, SAM, trained on one billion masks, demonstrates exceptional generalization ability, inspiring our research. However, using SAM for iimage manipulation detection tasks presents significant challenges: a) reliance on manual prompts: SAM's interactive framework requires predefined prompts for input images, such as points, boxes, or masks. As a category-agnostic segmentation method, these limitations hinder SAM's ability to achieve fully automated understanding in IMD tasks; b) single-view information struggles to support cross-dataset generalization: previous studies Zhou et al. (2020) have reported that DeepLabv2 Chen et al. (2017) trained on the CASIAv2 Dong et al. (2013) dataset performed well on the CAISAv1 Dong et al. (2013) dataset (sourced from CASIAv2) but poorly on the non-sourced COVER Wen et al. (2016) dataset. Similar behavior has also been observed with FCN Long et al. (2015) in this research.

To activate SAM's IMD capabilities, we propose a cross-view automated prompt learning paradigm called IMDPrompter, aimed at learning how to generate prompts to enhance SAM's functionality: a) Inspired by Battle & Gollapudi (2024) using automated prompt optimizers to improve the performance of LLMs, we constructed the automated prompt learning paradigm IMDPrompter, which generates prompt information automatically; b) To enhance SAM's cross-dataset generalization ability, we introduced three noise views: SRM filtering, Bayer Conv, and Noiseprint, aiming to generate enhanced prompt information from multiple perspectives. To integrate information from multiple views, previous methods Zhai et al. (2023) employed a strategy of weighted averaging of the outputs from multiple views. However, due to the inaccuracy of prompts generated from individual views, simple weighted averaging can smooth out the accurate prompts from the optimal view to suboptimal ones. To

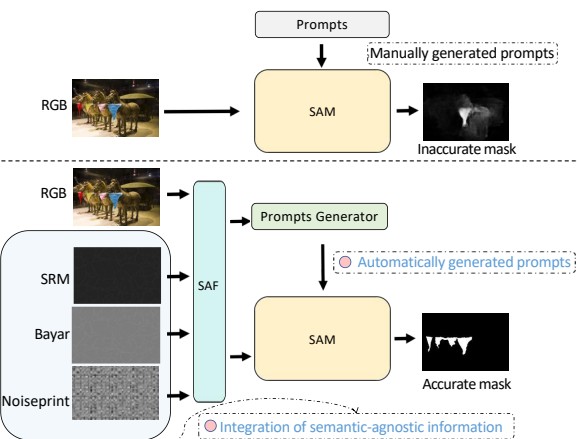

Figure 1: Improvements of proposed IMDPrompter over existing pipelines of baselines: (a). Automated prompt learning without the need for manual input. (b). Flexible integration of semantic-agnostic information crucial for Image Manipulation Detection.

preserve the accurate prompts from the optimal view, we propose an Optimal Prompt Selection (OPS) module based on minimizing localization loss; meanwhile, we noted that ideally, the prompt information from each view should converge toward the optimal prompt. Based on this motivation, we constructed a Cross-View Prompt Consistency (CPC) constraint to achieve alignment enhancement across views. For the mask generation process, we developed an Attention-based Cross-view Feature Perception (CFP) module and a Multi-Layer Perceptron-based Prompt Mixing module (PMM) to achieve the fusion of cross-view information and the integration of multiple types of prompts sequentially.

In summary, our main contributions are as follows.

- We are the first to apply SAM to the field of image manipulation detection and propose an automated prompt learning paradigm, IMDPrompter, eliminating the original SAM's reliance on manual prompts.

- We propose modules such as Optimal Prompt Selection and Cross-View Prompt Consistency Constraint, achieving alignment enhancement across views.

- We propose Cross-view Feature Perception and Prompt Mixing modules, achieving the fusion of cross-view information and the integration of multiple types of prompts.

- We demonstrate extensive results across five different image manipulation detection datasets, thoroughly validating the strong in-distribution and out-of-distribution image manipulation detection and localization capabilities of IMDPrompter.

## 2 RELATED WORK

**Image Manipulation Detection.** Currently, methods for image manipulation detection can be broadly categorized into two types, primarily distinguished by their recognition of manipulated artifacts. Some techniques Wu et al. (2019); Chen et al. (2021); Wu et al. (2022); Bi et al. (2019); Hu et al. (2020); Yang et al. (2020); Marra et al. (2020) rely on detecting abnormal features and often use high-pass noise filters Yang et al. (2020); Li & Huang (2019) to suppress content information. Other methods Park et al. (2018); Kwon et al. (2022); Mareen et al. (2022) attempt to detect inconsistencies in compression within tampered images, as they assume different compression Quality Factors (QFs) before and after the operation. Additionally, some researchers focus their attention on camera-based artifacts, such as model fingerprints Mareen et al. (2022); Cozzolino & Verdoliva (2019); Cozzolino et al. (2015).

**Foundational Models.** In recent years, foundational models have sparked a tremendous transformation in the field of artificial intelligence . These models, trained on extensive datasets, have demonstrated impressive generalization capabilities across various scenarios Kirillov et al. (2023); Cai et al. (2024); Liu et al. (2024b); Pan et al. (2024b;a; 2025). Renowned models such as Chat-GPT Ouyang et al. (2022), GPT-4 Achiam et al. (2023), and Stable Diffusion Rombach et al. (2022) have further propelled the development of artificial intelligence, making significant contributions to human civilization and exerting considerable influence across various industries. Inspired by the success of foundational models in natural language processing (NLP), researchers have begun exploring their potential applications in computer vision. While most of these models are aimed at extracting accessible knowledge from freely available data Alayrac et al. (2022); Radford et al. (2021); Chen et al. (2023), the recent SAM model Kirillov et al. (2023) adopts an innovative approach by constructing a data engine where the model co-develops annotations with environmental datasets. SAM uniquely leverages a vast collection of masks, showcasing robust generalization capabilities. However, it was initially designed as a task-agnostic segmentation model, requiring prompts (i.e., inputs of prior points, bounding boxes, or masks), and therefore does not directly facilitate end-to-end automated segmentation perception. This paper does not delve into the design and training of foundational image manipulation detection models; instead, we explore the potential of utilizing SAM's powerful universal segmentation capabilities for image manipulation detection and localization. Furthermore, the proposed method of learning prompts can be extended to other visual foundational models beyond SAM.

**Prompt Learning.** In the past, machine learning tasks were primarily focused on fully supervised learning, where task-specific models were trained only on labeled instances of the target task Krizhevsky et al. (2017). However, over time, there has been a significant shift in learning paradigms, transitioning from fully supervised learning towards *pretraining and fine-tuning* approaches for downstream tasks. This shift allows models to leverage general features acquired during pretraining Russakovsky et al. (2015); Simonyan & Zisserman (2014); He et al. (2016). More recently, with the advent of foundational models, a new paradigm has emerged known as *pretraining and prompting* Chen et al. (2023); Lester et al. (2021); Liu et al. (2023); Zhou et al. (2022). In this paradigm, researchers no longer train models specifically for downstream tasks but instead redesign inputs using prompts to reformulate the downstream tasks to align with the original pretraining task Radford et al. (2021); Devlin et al. (2018); Radford et al. (2019). Prompting helps to reduce semantic gaps, bridge the gap between pretraining and fine-tuning, and prevent overfitting of the heads. Since the advent of GPT-3 Brown et al. (2020), prompting has evolved from traditional discrete Liu et al. (2023) and continuous prompt constructions Chen et al. (2023); Zhou et al. (2022) to large-scale model-centric contextual learning Alayrac et al. (2022), instruction tuning Liu et al. (2024a); Peng et al. (2023); Gupta et al. (2022), and chaining approaches Wei et al. (2022); Wang et al. (2022a); Zhang et al. (2022). Currently, methods for constructing prompts include manual templates, heuristic-based templates, generation, fine-tuning word embeddings, and pseudo-labeling Liu et al. (2023); Wang et al. (2022b). In this paper, we propose a prompt generator for generating prompts compatible with SAM.

# 3 PROPOSED METHOD

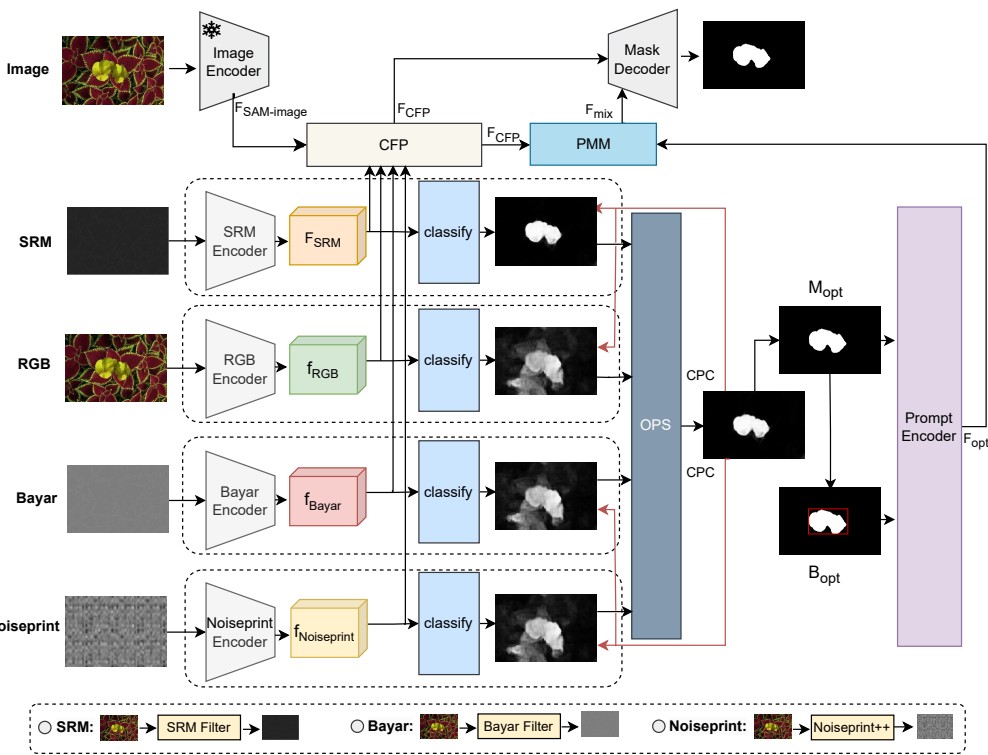

Figure 2: Overall framework of IMDPrompter. The prompter part consists of four views: RGB, SRM, Bayar and Noiseprint. OPS selects the optimal prediction from the four views to generate the best prompt. CPC enhances cross-view consistency. CFP achieves cross-view feature perception fusion. PMM achieves a mixture of multiple prompt information.

As illustrated in Figure 2, we introduce IMDPrompter, an automated prompt learning framework leveraging cross-view perception to unlock SAM's detection and localization capabilities. It employs an optimal prompt selection module to identify the best prompts, enhances alignment via cross-view prompt consistency, and achieves efficient image manipulation detection and localization through feature fusion and prompt mixing.

## 3.1 MULTI-VIEW FEATURE REPRESENTATION

Current technology Zhou et al. (2020) shows that while RGB views work well for in-domain (IND) manipulation detection, they perform poorly in out-of-domain (OOD) detection. Moreover, using noise views to capture semantic-agnostic information enhances performance. Therefore, relying solely on RGB view data is inadequate for detecting and localizing manipulations. To address this, we incorporate three semantic-agnostic views: SRM, Bayer, and Noiseprint noise views, as outlined below:

$$F_{\text{sam}} = \Phi_{\text{sam-img}}(\mathcal{I}), \tag{1}$$

$$f_{\text{RGB}} = \Phi_{\text{Seg-RGB}}(\mathcal{I}), f_{\text{SRM}} = \Phi_{\text{Seg-SRM}}\left(\Phi_{\text{SRM}}(\mathcal{I})\right), f_{\text{Bayer}} = \Phi_{\text{Seg-Bayer}}\left(\Phi_{\text{Bayer}}(\mathcal{I})\right), \tag{2}$$

$$f_{\text{Noiseprint}} = \Phi_{\text{Seg-Noiseprint}}\left(\Phi_{\text{Noiseprint}}(\mathcal{I})\right). \tag{3}$$

In the system, $\mathcal{I}$ represents the input image, and $\Phi_{\text{sam-img}}$ denotes the SAM image encoder. $F_{\text{sam}}$ represents the features encoded by the SAM image encoder. $\Phi_{\text{SRM}}$, $\Phi_{\text{Bayer}}$ and $\Phi_{\text{Noiseprint}}$ respectively represent the SRM noise map extractor, the Bayer noise map extractor and the Noiseprint noise map extractor. $\Phi_{\text{Seg-RGB}}$, $\Phi_{\text{Seg-SRM}}$, $\Phi_{\text{Seg-Bayer}}$ and $\Phi_{\text{Seg-Noiseprint}}$ each represent the segmenter for the RGB view, SRM view, Bayer view and Noiseprint view, with all four segmenters having identical structures

but non-shared parameters. $f_{\text{RGB}}$, $f_{\text{SRM}}$, $f_{\text{Bayer}}$ and $f_{\text{Noiseprint}}$ represent the features from the RGB, SRM, Bayer, and Noiseprint views, respectively.

## 3.2 OPTIMAL PROMPT SELECTION

In the masked prompt generation process, we integrate mask probability distributions from four views: $P_{\text{RGB}}$, $P_{\text{SRM}}$, $P_{\text{Bayar}}$, and $P_{\text{Noiseprint}}$, to form an ensemble distribution $P_{\text{Ens}}$. We select the optimal mask probability segmentation $P_{\text{opt}}$ by minimizing segmentation loss, and derive the corresponding mask $M_{\text{opt}}$ and bounding box prompts $\mathcal{B}_{\text{opt}}$. These are fed into SAM's prompt encoder to obtain the prompt encoding $F_{\text{opt}}$. The process is outlined as follows:

$$P_{\text{RGB}} = \Phi_{\text{RGB}-\text{CLS}}(f_{\text{RGB}}), P_{\text{SRM}} = \Phi_{\text{SRM}-\text{CLS}}(f_{\text{SRM}}), P_{\text{Bayer}} = \Phi_{\text{Bayer}-\text{CLS}}(f_{\text{Bayer}}), \quad (4)$$

$$P_{\text{Noiseprint}} = \Phi_{\text{Noiseprint}-\text{CLS}}(f_{\text{Noiseprint}}), \quad (5)$$

$$P_{\text{Ens}} = \frac{P_{\text{RGB}} + P_{\text{SRM}} + P_{\text{Bayer}} + P_{\text{Noiseprint}}}{4}, \quad (6)$$

$$P_{\text{opt}} = \underset{P \in \mathcal{P}}{\arg\min} \mathcal{L}_{\text{Seg}}(P, G), \mathcal{P} = \{P_{\text{RGB}}, P_{\text{SRM}}, P_{\text{Bayer}}, P_{\text{Noiseprint}}, P_{\text{Ens}}\}, \quad (7)$$

$$M_{\text{opt}} = \Phi_{\text{mask}}(P_{\text{opt}}), \mathcal{B}_{\text{opt}} = \Phi_{\text{box}}(M_{\text{opt}}), \quad (8)$$

$$F_{\text{opt}} = \Phi_{\text{p}-\text{enc}}(M_{\text{opt}}, \mathcal{B}_{\text{opt}}), \quad (9)$$

where $\Phi_{\text{RGB}-\text{CLS}}$, $\Phi_{\text{SRM}-\text{CLS}}$, $\Phi_{\text{Bayer}-\text{CLS}}$ and $\Phi_{\text{Noiseprint}-\text{CLS}}$ represent classifiers for the RGB, SRM, Bayer, and Noiseprint views, respectively, $G$ represents the one-hot encoded mask labels, $\mathcal{L}_{\text{Seg}}$ is the segmentation loss function, $\Phi_{\text{mask}}$ represents the mask generation operation, $\Phi_{\text{box}}$ is the bounding box generation operation, and $\Phi_{\text{p}-\text{enc}}$ represents SAM's prompt encoder.

## 3.3 CROSS-VIEW PROMPT CONSISTENCY

Ideally, the segmentation masks from the four views should be consistent with the optimal segmentation mask. Therefore, we constructed a cross-view prompt consistency enhancement loss to achieve enhanced prompt consistency across views. The CPC loss function is expressed as follows:

$$\mathcal{L}_{\text{CPC}} = \mathcal{L}_{\text{Seg}}(P_{\text{RGB}}, P_{\text{opt}}) + \mathcal{L}_{\text{Seg}}(P_{\text{SRM}}, P_{\text{opt}}) + \mathcal{L}_{\text{Seg}}(P_{\text{Bayer}}, P_{\text{opt}}) + \mathcal{L}_{\text{Seg}}(P_{\text{Noiseprint}}, P_{\text{opt}}), \quad (10)$$

where $\mathcal{L}_{\text{Seg}}$ is the segmentation loss function.

## 3.4 CROSS-VIEW FEATURE PERCEPTION

To enhance the segmentation accuracy of SAM, we propose a Cross-View Feature Perception (CFP) module that integrates features from RGB, SRM, Bayer, and Noiseprint views. This module enriches the feature representations for SAM's decoding process, as shown in Figure 3. Here, $\Phi_{\text{CFP}}$ denotes the CFP module, and $F_{\text{CFP}}$ represents the fused cross-view features.

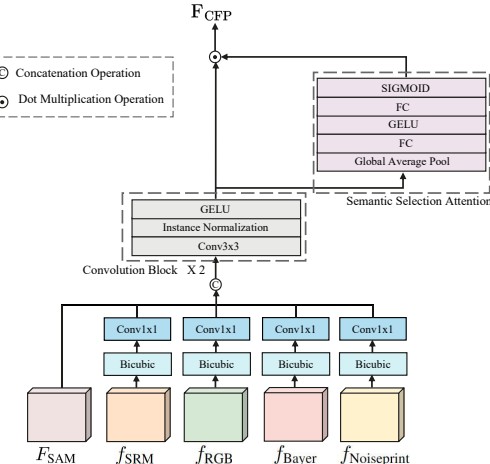

The process begins by aligning features $f_{\text{RGB}}$, $f_{\text{SRM}}$, $f_{\text{Bayer}}$, and $f_{\text{Noiseprint}}$ with $F_{\text{sam}}$ via bicubic interpolation. These features are then concatenated with $F_{\text{sam}}$ after a $1 \times 1$ convolution. The concatenated features pass through two stacked convolution blocks ($3 \times 3$ convolution, instance normalization, and GELU activation). Meanwhile, the semantic selection attention branch (global average pooling, fully connected layer, GELU, and sigmoid) weights the shortcut branch to produce the integrated feature $F_{\text{CFP}}$. The CFP module effectively fuses multi-view features to improve segmentation performance.

Figure 3: Architecture of the Cross-view Feature Perception (CFP) unit.

$$F_{\text{CFP}} = \Phi_{\text{CFP}}(F_{\text{sam}}, f_{\text{RGB}}, f_{\text{SRM}}, f_{\text{Bayer}}, f_{\text{Noiseprint}}) \quad (11)$$

## 3.5 PROMPT MIXING MODULE

Since $F_{\text{CFP}}$ aggregates semantically related and unrelated features, it aids the guidance of SAM for image manipulation detection. Therefore, we constructed the PMM module, which is primarily based on MLP, to integrate multiple types of prompt information. First, the two types of prompt inputs, $F_{\text{CFP}}$ and $F_{\text{opt}}$, are concatenated, followed by an MLP layer to perform dimension transformation to align with the original SAM prompt encoder's prompt embedding. Then, $F_{\text{mix}}$ and $F_{\text{CFP}}$ are input into SAM's mask decoder to achieve the image manipulation localization process. The process is as follows:

$$F_{\text{mix}} = \Phi_{\text{MLP}}\left([F_{\text{CFP}}; F_{\text{opt}}]\right), \tag{12}$$

$$P_{\text{sam}} = \Phi_{\text{sam}-\text{dec}}(F_{\text{CFP}}, F_{\text{mix}}), \tag{13}$$

where $\Phi_{\text{MLP}}$ represents the MLP layer operation, $F_{\text{mix}}$ represents the integrated prompt embedding, and $\Phi_{\text{sam}-\text{dec}}$ represents SAM's mask decoder, with $P_{\text{sam}}$ representing the mask probability prediction output of the mask decoder.

## 3.6 TRAINING AND INFERENCE PIPELINE

The training loss function of IMDPrompter includes the segmentation loss from four prompt views $\mathcal{L}_{\text{Seg-p}}$, the CPC loss $\mathcal{L}_{\text{CPC}}$, the SAM decoder segmentation loss $\mathcal{L}_{\text{Seg-sam}}$, and the image-level prediction loss $\mathcal{L}_{\text{Img-level}}$. The formula is expressed as follows:

$$\mathcal{L} = \mathcal{L}_{\text{Seg-sam}} + \lambda_1 \mathcal{L}_{\text{Seg-p}} + \lambda_2 \mathcal{L}_{\text{CPC}} + \lambda_3 \mathcal{L}_{\text{Img-level}}, \tag{14}$$

where

$$L_{\text{Seg-sam}} = L_{\text{Seg}}(P_{\text{sam}}, G), \tag{15}$$

$$L_{\text{Seg-p}} = L_{\text{Seg}}(P_{\text{RGB}}, G) + L_{\text{Seg}}(P_{\text{Bayar}}, G) + L_{\text{Seg}}(P_{\text{SRM}}, G) + L_{\text{Seg}}(P_{\text{Noiseprint}}, G). \tag{16}$$

In this paper, we use Focal Loss as our $L_{\text{Seg}}$.

For image-level detection, following the work of Zhai et al. (2023), we adopt an adaptive pooling based on minimizing intra-class prediction variance. The overall computational process is as follows, first using Otsu's method Otsu et al. (1975) to find a threshold $\omega_0$ that minimizes the intra-class prediction variance:

$$\omega_0 = \underset{\omega \in \{\hat{p}_{i,j}\}}{\arg\min} |\{\hat{p}_{i,j} \mid \hat{p}_{i,j} < \omega\}| \operatorname{var}\left(\{\hat{p}_{i,j} \mid \hat{p}_{i,j} < \omega\}\right) +$$
$$|\{\hat{p}_{i,j} \mid \hat{p}_{i,j} \geq \omega\}| \operatorname{var}\left(\{\hat{p}_{i,j} \mid \hat{p}_{i,j} \geq \omega\}\right), \tag{17}$$

where $\operatorname{var}(\cdot)$ denotes variance, $\hat{p}_{i,j}$ is the pixel-level response at position $(i, j)$. Then the image-level prediction is aggregated from pixel-level responses above the threshold and the image-level loss is:

$$\hat{y}_{\text{A}} = \frac{1}{|\mathbb{P}_{\text{h}}|} \sum_{\hat{p} \in \mathbb{P}_{\text{h}}} \hat{p}; \mathbb{P}_{\text{h}} = \{\hat{p}_{i,j} \mid \hat{p}_{i,j} \geq \omega_0\}, \tag{18}$$

$$\mathcal{L}_{\text{Img-level}} = \mathcal{L}_{\text{BCE}}(y, \hat{y}_{\text{A}}), \tag{19}$$

where $\hat{y}_{\text{A}}$ is the image-level prediction, $\mathcal{L}_{\text{BCE}}$ denotes the binary cross-entropy loss.

During the inference process of IMDPrompter, since there are no true labels, our OPS module defaults to selecting $P_{\text{Ens}}$ for generating masks and bounding box prompts, with other components functioning as during the training process.

## 4 EXPERIMENT

**Dataset.** Our method is trained only on the CASIAv2 dataset Dong et al. (2013). For in-distribution (IND) evaluation, we use the CASIAv1 dataset Dong et al. (2013). For out-of-distribution (OOD) evaluation, we use three datasets: Columbia Hsu & Chang (2006), Coverage Wen et al. (2016)and IMD2020 Novozamsky et al. (2020).

**Evaluation Metrics.** For image-level manipulation detection, we report specificity, sensitivity, and their F1-score (I-F1). The area under the receiver operating characteristic curve (AUC) is also

reported as a threshold-independent metric for image-level detection. For pixel-level manipulation localization, we follow previous methods Chen et al. (2021); Zhou et al. (2018b;a); Salloum et al. (2018) to compute pixel accuracy, recall, and their F1-score (P-F1) on manipulated images. The overall performance of image and pixel-level manipulation detection/localization is measured by the harmonic mean of pixel-level and image-level F1-scores Chen et al. (2021), denoted as composite F1 (C-F1), and is sensitive to lower values of P-F1 and I-F1. To ensure fair comparison, a default threshold of 0.5 is used for F1 computation unless otherwise specified.

**Implementation Details.** In our experiments, unless otherwise specified, we consistently use the VIT-L backbone of SAM and employ FCN (lightweight architecture based on MobileNet Howard (2017)) as the segmentor for the three prompt views. We maintain image size at 1024×1024, consistent with the original input of the SAM model. To augment training samples, we use data augmentation techniques such as horizontal flipping and random cropping. The image encoder remains frozen during the training phase. All experiments are run on NVIDIA A6000 GPUs. For the optimization process, we train our model using the AdamW optimizer with an initial learning rate of 1e-4. We use a batch size of 4 and train for 100 epochs. We implement a linear warm-up strategy with a cosine annealing scheduler Loshchilov & Hutter (2016) to decay the learning rate.

## 4.1 COMPARISON WITH STATE-OF-THE-ART METHOD

Table 1: Pixel-level manipulation detection performance. Performance metric is F1-score (%). The best results for each test set are highlighted in bold and second-best values are underlined.

| Method | Best threshold | | | | | | Fixed threshold (0.5) | | | | | |
|---|---|---|---|---|---|---|---|---|---|---|---|---|
| | CASIA | COVER | Columbia | IMD | NIST | MEAN | CASIA | COVER | Columbia | IMD | NIST | MEAN |
| MFCN Salloum et al. (2018) | 54.1 | - | 61.2 | - | - | - | - | - | - | - | - | - |
| RGB-N Zhou et al. (2018b) | 40.8 | 37.9 | - | - | - | - | - | - | - | - | - | - |
| H-LSTM Bappy et al. (2019) | 20.9 | 21.3 | 14.2 | 31.0 | 46.6 | 26.80 | 15.4 | 16.3 | 13.0 | 19.5 | 35.4 | 19.92 |
| ManTra-Net Wu et al. (2019) | 69.2 | 77.2 | 70.9 | 70.5 | 45.5 | 66.66 | 15.5 | 28.6 | 36.4 | 18.7 | 0.0 | 19.84 |
| HP-FCN Li & Huang (2019) | 21.4 | 19.9 | 47.1 | 16.9 | 36.0 | 28.26 | 15.4 | 0.3 | 6.7 | 11.2 | 12.1 | 9.14 |
| CR-CNN Yang et al. (2020) | 66.2 | 47.0 | 70.4 | 60.0 | 42.8 | 57.28 | 40.5 | 29.1 | 43.6 | 26.2 | 23.8 | 32.64 |
| GSR-Net Zhou et al. (2018a) | 57.4 | 48.9 | 62.2 | 68.7 | 45.6 | 56.56 | 38.7 | 28.5 | 61.3 | 24.3 | 28.3 | 36.22 |
| SPAN Hu et al. (2020) | 68.8 | 71.8 | 77.4 | 69.6 | 68.3 | 71.18 | 18.4 | 17.2 | 48.7 | 17.0 | 22.1 | 24.68 |
| CAT-Net Kwon et al. (2021) | 57.3 | 48.5 | 77.6 | 51.7 | 59.9 | 59.00 | 13.6 | 12.9 | 55.5 | 5.4 | 17.9 | 21.06 |
| MVSS-Net Chen et al. (2021) | 75.3 | 82.4 | 70.3 | 75.7 | 73.7 | 75.48 | 45.2 | 45.3 | 63.8 | 26.0 | 29.2 | 41.90 |
| Trufor Guillaro et al. (2023) | 83.5 | 74.1 | 82.1 | - | 68.8 | - | 73.7 | 60.0 | 85.9 | - | 39.9 | - |
| FCN Long et al. (2015) | 74.2 | 57.3 | 58.6 | 64.5 | 50.7 | 61.06 | 44.1 | 19.9 | 22.3 | 21.0 | 16.7 | 24.80 |
| IMDPrompter | **85.1** | **83.5** | **87.3** | **76.3** | **74.6** | **81.36** | **76.3** | **63.6** | **87.3** | **30.6** | **41.1** | **59.78** |
| | +10.9 | +26.2 | +28.7 | +11.8 | +23.9 | +20.3 | +32.2 | +43.7 | +65.0 | +9.6 | +24.4 | +34.98 |

Table 2: Image-level manipulation detection performance. Decision threshold: 0.5. NIST16 is excluded as it lacks genuine images. The best results for each test set are highlighted in bold and second-best values are underlined.

| Method | CASIA | | | | COVER | | | | Columbia | | | | IMD | | | | MEAN | |
|---|---|---|---|---|---|---|---|---|---|---|---|---|---|---|---|---|---|---|
| | AUC | Sen. | Spe. | F1 | AUC | Sen. | Spe. | F1 | AUC | Sen. | Spe. | F1 | AUC | Sen. | Spe. | F1 | AUC | F1 |
| H-LSTM Bappy et al. (2019) | 0.498 | 99.7 | 0.0 | 0.0 | 0.500 | 100.0 | 0.0 | 0.0 | 0.506 | 100.0 | 1.1 | 2.2 | 0.501 | 100.0 | 0.0 | 0.0 | 0.5013 | 0.55 |
| ManTra-Net Wu et al. (2019) | 0.500 | 100.0 | 0.0 | 0.0 | 0.500 | 100.0 | 0.0 | 0.0 | 0.701 | 100.0 | 0.0 | 0.0 | 0.500 | 100.0 | 0.0 | 0.0 | 0.5503 | 0.00 |
| CR-CNN Yang et al. (2020) | 0.719 | 93.0 | 13.9 | 24.2 | 0.566 | 96.7 | 7.0 | 13.1 | 0.783 | 96.1 | 24.6 | 39.2 | 0.615 | 92.9 | 12.3 | 21.7 | 0.6708 | 24.55 |
| GSR-Net Zhou et al. (2018a) | 0.500 | 99.4 | 0.0 | 0.0 | 0.515 | 100.0 | 0.0 | 0.0 | 0.502 | 100.0 | 1.1 | 2.2 | 0.500 | 100.0 | 0.0 | 0.0 | 0.5043 | 0.55 |
| SPAN Hu et al. (2020) | 0.500 | 100.0 | 0.0 | 0.0 | 0.500 | 100.0 | 0.0 | 0.0 | 0.500 | 100.0 | 0.0 | 0.0 | 0.500 | 100.0 | 0.0 | 0.0 | 0.5000 | 0.00 |
| CAT-Net Kwon et al. (2021) | 0.647 | 23.9 | 92.1 | 38.0 | 0.557 | 28.0 | 80.0 | 41.5 | 0.971 | 87.2 | 96.2 | 91.5 | 0.586 | 27.5 | 81.6 | 41.1 | 0.6903 | 53.03 |
| MVSS-Net Chen et al. (2021) | 0.937 | 61.5 | 98.8 | 75.8 | 0.731 | 94.0 | 14.0 | 24.4 | 0.980 | 66.9 | 100.0 | 80.2 | 0.656 | 91.5 | 22.0 | 35.5 | 0.8260 | 53.98 |
| Trufor Guillaro et al. (2023) | 0.916 | - | - | - | 0.770 | - | - | - | **0.996** | - | - | - | - | - | - | - | - | - |
| FCN Long et al. (2015) | 0.770 | 72.8 | 64.3 | 68.3 | 0.541 | 90.0 | 10.0 | 18.0 | 0.762 | 95.0 | 32.2 | 48.1 | 0.502 | 84.6 | 15.5 | 26.2 | 0.6438 | 40.15 |
| IMDPrompter | **0.978** | 91.6 | 66.9 | **77.3** | **0.796** | 96.8 | 55.2 | **70.3** | 0.983 | 97.3 | 90.2 | **93.6** | **0.671** | 71.6 | 57.4 | **63.7** | **0.8570** | **76.23** |
| | +0.208 | +18.8 | +2.6 | +9.0 | +0.255 | +6.8 | +45.2 | +52.3 | +0.221 | +2.3 | +58.0 | +45.5 | +0.169 | -13.0 | +41.9 | +37.5 | +0.2132 | +36.08 |

**Pixel-level Manipulation Detection** Table 1 illustrates the pixel-level detection performance of different models. We evaluate the F1-score under two settings: best threshold and fixed threshold (0.5). From Table 1, it is evident that IMDPrompter achieves the best performance across nearly all datasets. Under the optimal threshold setting, we achieved an average F1 Score of 81.36%. Under the fixed threshold setting, IMDPrompter achieved an average F1 Score of 59.78%, indicating that our proposed IMDPrompter demonstrates better robustness in threshold settings.

**Image-level Manipulation Detection.** Table 2 presents the image-level performance of different models. For image-level performance, we use a default decision threshold of 0.5. Once again, IMDPrompter emerges as the top performer, leveraging our prompt learning paradigm to achieve

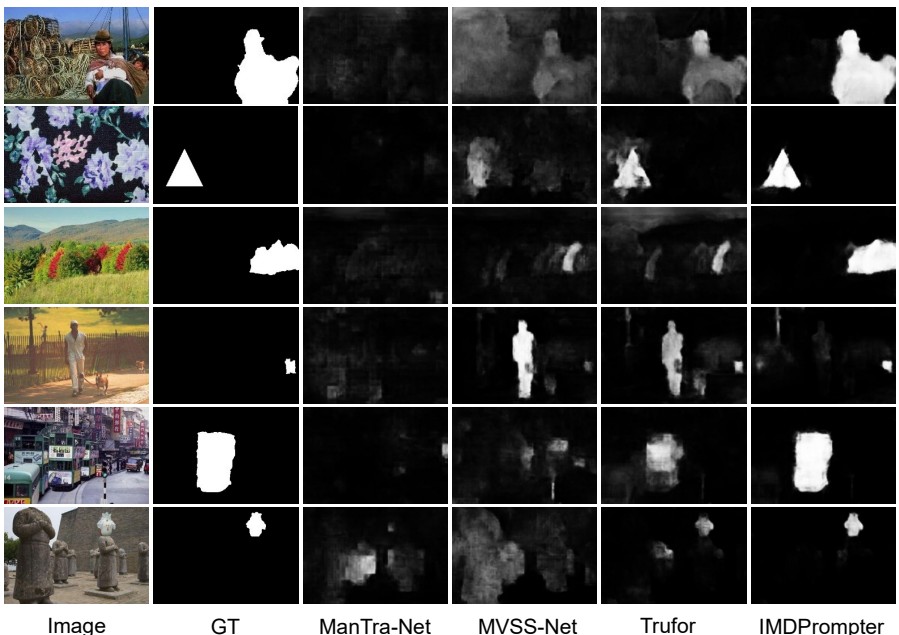

Figure 4: Visualization Comparison of Image Manipulation Detection and Localization Results

Table 3: Evaluation of each component of our method in IND dataset.

|  | CFP | CPC | OPS | PMM | CASIA I-F1 | CASIA P-F1 | CASIA C-F1 |
|---|---|---|---|---|---|---|---|
| 1 |  |  |  |  | 70.6 | 70.3 | 70.4 |
| 2 | ✓ |  |  |  | 73.2 | 73.1 | 73.1 |
| 3 | ✓ | ✓ |  |  | 74.3 | 74.3 | 74.3 |
| 4 | ✓ | ✓ | ✓ |  | 75.1 | 76.1 | 75.6 |
| 5 | ✓ | ✓ | ✓ | ✓ | 77.3 | 76.3 | 76.8 |

Table 4: Evaluation of each component of our method in OOD dataset.

|  | CFP | CPC | OPS | PMM | COVER I-F1 | COVER P-F1 | COVER C-F1 |
|---|---|---|---|---|---|---|---|
| 6 |  |  |  |  | 20.4 | 39.8 | 27.0 |
| 7 | ✓ |  |  |  | 49.6 | 54.3 | 51.8 |
| 8 | ✓ | ✓ |  |  | 58.6 | 57.0 | 57.8 |
| 9 | ✓ | ✓ | ✓ |  | 65.2 | 61.4 | 63.2 |
| 10 | ✓ | ✓ | ✓ | ✓ | 70.3 | 63.6 | 66.8 |

higher specificity in most test settings, thereby reducing false positives. Additionally, IMDPrompter achieves the best average AUC and average F1-scores. The average F1-score of IMDPrompter significantly surpasses that of the second-best method, MVSS-Net, with an improvement of 22.25%.

**Qualitative Results.** As shown in Figure 10, we visualized the image manipulation localization results of four methods: ManTra-Net, MVSS-Net, Trufor, and IMDPrompter. It is evident that IMDPrompter generates more accurate confidence maps for image manipulation localization, achieving better detection and localization.

## 4.2 ABLATION STUDIES

To reveal the impact of different components, we evaluated the proposed model under various settings, analyzing the distinct effects of each component on in-domain and out-of-domain datasets. As shown in Tables 3 and 4, we conducted ablation experiments on the in-domain dataset CASIA and the out-of-domain dataset COVER, respectively.

**Baseline.** From *Experiment 1*, it is known that our baseline method achieved performance scores of 70.6%, 70.3%, and 70.4% on the I-F1, P-F1, and C-F1 metrics respectively in the CASIA dataset. In the COVER dataset, it scored 20.4%, 39.8%, and 27.0% on the same metrics. These experiments show that using only RGB visual view information has limited generalization ability in out-of-domain datasets.

**Impact of CFP.** From *Experiment 2*, By implementing cross-view feature perception fusion through the CFP module, the I-F1, P-F1, and C-F1 metrics on the CASIA dataset improved by 2.6%, 2.8%,

Table 5: Ablation Study of View Combinations

| RGB | Noiseprint | Bayar | SRM | CASIA | | | | COVER | | | | Columbia | | | | IMD | | | |
|---|---|---|---|---|---|---|---|---|---|---|---|---|---|---|---|---|---|---|---|
| | | | | I-AUC | I-F1 | P-F1 | C-F1 | I-AUC | I-F1 | P-F1 | C-F1 | I-AUC | I-F1 | P-F1 | C-F1 | I-AUC | I-F1 | P-F1 | C-F1 |
| ✓ | | | | 0.856 | 73.60 | 69.60 | 71.50 | 0.556 | 19.60 | 38.60 | 26.00 | 0.786 | 53.60 | 24.90 | 34.00 | 0.531 | 28.10 | 21.20 | 24.20 |
| | ✓ | | | 0.903 | 75.90 | 73.60 | 74.70 | 0.761 | 62.90 | 58.70 | 60.70 | 0.946 | 88.60 | 83.50 | 86.00 | 0.636 | 59.90 | 27.90 | 38.10 |
| | | ✓ | | 0.863 | 74.30 | 70.60 | 72.40 | 0.706 | 61.30 | 56.30 | 58.70 | 0.889 | 83.40 | 81.60 | 82.50 | 0.601 | 54.60 | 25.70 | 34.90 |
| | | | ✓ | 0.896 | 75.10 | 71.60 | 73.30 | 0.713 | 59.60 | 55.60 | 57.50 | 0.906 | 84.60 | 79.30 | 81.90 | 0.593 | 56.30 | 24.30 | 33.90 |
| ✓ | ✓ | | | 0.941 | 76.90 | 75.20 | 76.00 | 0.781 | 66.80 | 61.90 | 64.30 | 0.969 | 91.70 | 85.40 | 88.40 | 0.655 | 61.50 | 28.80 | 39.20 |
| ✓ | ✓ | ✓ | | 0.962 | 77.10 | 75.90 | 76.50 | 0.788 | 68.70 | 62.40 | 65.40 | 0.977 | 92.40 | 86.50 | 89.40 | 0.660 | 62.60 | 29.50 | 40.10 |
| ✓ | ✓ | ✓ | ✓ | 0.978 | 77.30 | 76.30 | 76.80 | 0.796 | 70.30 | 63.60 | 66.80 | 0.983 | 93.60 | 87.30 | 90.30 | 0.671 | 63.70 | 30.60 | 41.30 |

Table 6: Ablation Study of SAM

| Method | CASIA | | | | COVER | | | | Columbia | | | | IMD | | | |
|---|---|---|---|---|---|---|---|---|---|---|---|---|---|---|---|---|
| | I-AUC | I-F1 | P-F1 | C-F1 | I-AUC | I-F1 | P-F1 | C-F1 | I-AUC | I-F1 | P-F1 | C-F1 | I-AUC | I-F1 | P-F1 | C-F1 |
| FCN | 0.770 | 68.3 | 44.1 | 53.6 | 0.541 | 18.0 | 19.9 | 18.9 | 0.762 | 48.1 | 22.3 | 30.5 | 0.502 | 26.2 | 21.0 | 23.3 |
| FCN+ | 0.873 | 73.4 | 61.3 | 66.8 | 0.681 | 49.2 | 43.9 | 46.4 | 0.871 | 72.6 | 52.4 | 60.9 | 0.589 | 43.1 | 25.1 | 31.7 |
| IMDPrompter | 0.978 | 77.3 | 76.3 | 76.8 | 0.796 | 70.3 | 63.6 | 66.8 | 0.983 | 93.6 | 87.3 | 90.3 | 0.671 | 63.7 | 30.6 | 41.3 |

and 2.7%, respectively. Experiment VII showed that on the COVER dataset, the I-F1, P-F1, and C-F1 metrics increased by 29.2%, 14.5%, and 24.8%, respectively. This indicates that cross-view feature perception fusion can enhance performance on both in-domain and out-of-domain datasets, especially on out-of-domain datasets.

**Impact of CPC.** From *Experiment 3*, the inclusion of CPC, which introduced cross-view consistency enhancement, resulted in performance gains of 3.7%, 4.0%, and 3.9% on the I-F1, P-F1, C-F1 metrics respectively in the CASIA dataset. Experiment VIII showed performance gains of 38.2%, 17.2%, and 30.8% respectively on the COVER dataset, confirming that the prompt information between different views is complementary and integrative enhancement can bring objective performance improvements.

**Impact of OPS.** The inclusion of OPS for heuristic selection of optimal prompts resulted in improvements of 4.5%, 5.8%, and 5.2% in the I-F1, P-F1, and C-F1 metrics on the CASIA dataset, respectively. Experiment IX showed gains of 44.8%, 21.6%, and 36.2% respectively on the COVER dataset, indicating that simply adding together prompts from different views is not enough to generate the best prompts, and a optimal prompts selection strategy can optimize the prompt selection process. From *Experiment 4*, the inclusion of OPS adaptive selection of the optimal prompts resulted in performance gains of 4.5%, 5.8%, and 5.2% on the I-F1, P-F1, C-F1 metrics respectively in the CASIA dataset.

**Impact of PMM.** From *Experiment 5*, with the inclusion of PMM which fully integrates various types of prompt information, there was an increase in performance of 6.7%, 6.0%, and 6.4% on the I-F1, P-F1, C-F1 metrics respectively in the CASIA dataset. *Experiment 10* showed an increase of 49.9%, 23.8%, and 39.8% respectively on the COVER dataset, further demonstrating the effectiveness of PMM.

**Impact of View Combinations.** As shown in Table 5, the performance of using the RGB view alone does not differ significantly from that of using semantic-agnostic views such as SRM, Bayer, and Noiseprint individually on the in-domain test set CASIA. In contrast, for the out-of-domain test sets COVER, Columbia, and IMD, there is a significant performance gap between the RGB view and the semantic-agnostic views. This indicates that semantic-agnostic views play a crucial role in generalization on out-of-domain datasets. Additionally, we conducted ablation studies on various view combinations, and using both the RGB view and the Noiseprint view simultaneously resulted in substantial performance improvements. This demonstrates that the Noiseprint view serves as a critical semantic-agnostic feature. Next, we sequentially introduced the Bayer and SRM views, achieving further steady improvements across all metrics. When we combined the RGB, Noiseprint, SRM, and Bayer views, our method achieved optimal performance, validating the effectiveness of this combination.

**Impact of SAM.** As shown in Table 6, we constructed an FCN+ model that integrates four views without using SAM, further analyzing the impact of SAM on IMD. Due to the lack of potential priors from a large-scale pre-training dataset, the performance ceiling of FCN+ is relatively low. In contrast, our IMDPrompter fully leverages these potential priors, thereby enhancing the performance ceiling

**Impact of Quality Degradation.** As shown in Figure 5, following Dong et al. (2022), we evaluated the robustness of the models under two common image processing operations encountered during the

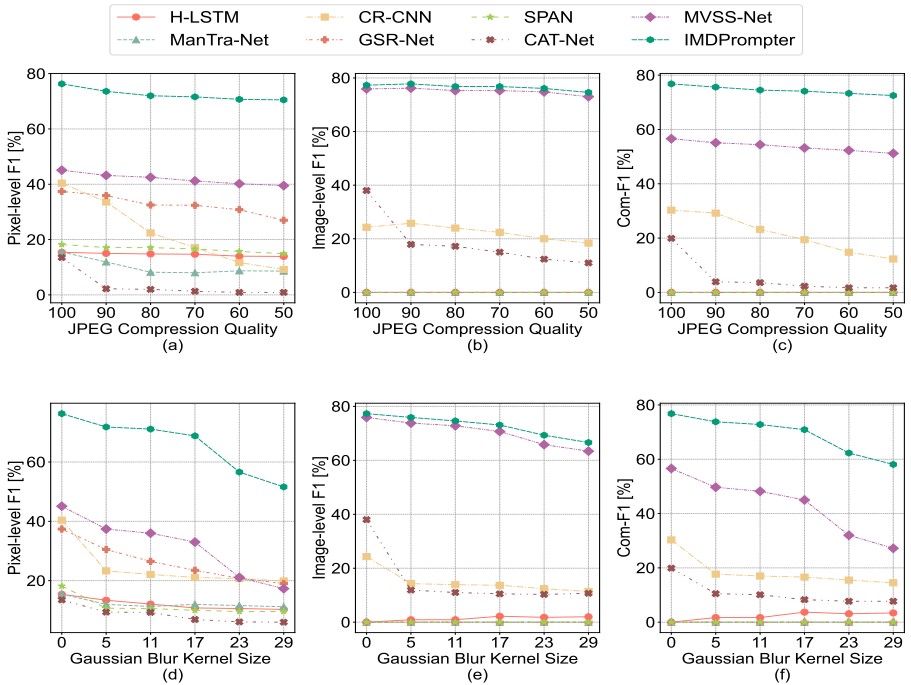

Figure 5: Robustness Analysis Against JPEG Compression and Gaussian Blur Interference

dissemination of images on the internet, namely JPEG compression and Gaussian blur. Comparing these two operations, Gaussian blur has a more significant impact on detection performance, especially when using larger 17x17 scale convolution kernels. Compared to previous methods, our IMDPrompter exhibits better robustness.

## 5 CONSLUSION

In this paper, we propose a cross-view perceptual prompt learning paradigm, IMDPrompter, which applies SAM to the image manipulation detection task for the first time. We sequentially introduce components such as optimal prompt selection, cross-view prompt consistency, cross-view feature perception, and prompt mixing modules to achieve efficient and automated image manipulation detection and localization. Our IMDPrompter demonstrates significant image-level and pixel-level manipulation detection performance in both in-domain (IND) and out-of-domain (OOD) scenarios, as well as across various robustness evaluation settings.

## 6 ACKNOWLEDGEMENTS

We thank all the anonymous reviewers for their helpful comments. This work was supported by the National Key R&D Program of China (2022YFB4701400/4701402), SSTIC Grant(KJZD20230923115106012,KJZD20230923114916032,GJHZ20240218113604008), Beijing Key Lab of Networked Multimedia and National Natural Science Foundation of China under Grant 62202302.

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

## 7 APPENDIX

### 7.1 LIMITATIONS AND BROADER IMPACTS

**Limitations.** We have some limitations in our method. First, it cannot detect completely generated images. Second, training IMDPrompter requires complete pixel-level supervision. In future research, we plan to explore more efficient label utilization through weakly supervised and semi-supervised learning approaches and extend these methods to newer foundational models such as SEEM Zou et al. (2024) and SAM2 Ravi et al. (2024).

**Broader Impacts.** In recent years, with the advancement of deep learning technologies Zhang et al. (2024; 2025b); Qiu et al. (2024; 2025b;a); Wu et al. (2025; 2024); Li et al. (2024); Hu et al. (2024); Wei et al. (2024b;a; 2025); Zhang & Qi (2024); Dong et al. (2023a;c;b), the editing and manipulation of digital media have become increasingly accessible and widespread. Advances in image editing software, along with deep generative models such as Generative Adversarial Networks and diffusion models Gao et al. (2024); Chen et al. (2024), have made image manipulations, often imperceptible to the human eye, much easier. These technologies are even being widely employed by potential malicious users. The ubiquitous use of smartphones and social networks has further accelerated the dissemination of these manipulated media. As a result, when these edited images are employed to support disinformation or to distort news content in order to mislead the public, they can create significant social problems and contribute to a crisis of trust. In this context, IMDPrompter, as an effective image manipulation detection method, can play a crucial role in mitigating the negative consequences of intentional image manipulation.

### 7.2 DATASET DESCRIPTION

In order to directly compare with state-of-the-art technologies, we trained on CASIAv2 Dong et al. (2013) and conducted extensive testing on COVER Wen et al. (2016), Columbia Hsu & Chang (2006), NIST16 Hsu & Chang (2006), CASIAv1 Dong et al. (2013), and the recent IMD Novozamsky et al. (2020).

Table 7: Details of the training set and five test sets used in our experiments. The symbol "–" indicates unavailable information. Copy-move, splicing, and inpainting operations are denoted as cpmv, spli, and inpa, respectively. Our model was trained on the CASIAv2 dataset and evaluated across all test sets.

| Dataset | Negative | Positive | cpmv | spli | inpa |
|---|---|---|---|---|---|
| **Training** | | | | | |
| CASIAv2 Dong et al. (2013) | 7491 | 5063 | 3235 | 1828 | 0 |
| **Testing** | | | | | |
| COVER Wen et al. (2016) | 100 | 100 | 100 | 0 | 0 |
| Columbia Hsu & Chang (2006) | 183 | 180 | 0 | 180 | 0 |
| NIST16 Dong et al. (2013) | 0 | 564 | 68 | 288 | 208 |
| CASIAv1+ Dong et al. (2013) | 800 | 920 | 459 | 461 | 0 |
| IMD Novozamsky et al. (2020) | 414 | 2010 | – | – | – |

### 7.3 BASELINE METHODS

As shown in Table 8, we have organized the technical routes of the baseline methods. To ensure fair and reproducible comparisons, we selected state-of-the-art models that meet any of the following criteria: 1) pretrained models released by the paper authors, 2) publicly available source code, and 3) adherence to a common evaluation protocol, where CASIAv2 is used for training and other public datasets for testing. Based on these criteria, we compiled a list of nine published baselines, as follows:

- H-LSTM Bappy et al. (2019): Pretrained on a custom dataset of 65k processed images and fine-tuned on NIST16 and IEEE Forensics Challenge data.

Table 8: Comparison of Technical Routes of Baseline Methods

| Method | View | | | Semantic Segmentation Backbone |
|---|---|---|---|---|
| | RGB | Noise | Fusion | |
| MFCNSalloum et al. (2018) | + | - | - | FCN |
| RGB-NZhou et al. (2018b) | + | SRM | Late Fusion(Bilinear Pooling) | Faster R-CNN |
| H-LSTMBappy et al. (2019) | + | - | - | Patch-LSTM |
| ManTra-NetWu et al. (2019) | + | SRM+Bayar | Early Fusion(Feature Concatenation) | Wider VGG |
| HP-FCNLi & Huang (2019) | - | High-pass filters | - | FCN |
| GSR-NetZhou et al. (2018a) | + | - | - | DeepLabv2 |
| CR-CNNYang et al. (2020) | - | Bayar | - | Mask R-CNN |
| SPANHu et al. (2020) | + | SRM+Bayar | Early Fusion(Feature Concatenation) | Wider VGG |
| MM-NetYang et al. (2021) | + | Bayar | Middle Fusion(Attention Guidance) | Mask R-CNN |
| JPEG-ComNetRao & Ni (2021) | + | SRM | Early Fusion(Feature Concatenation) | Siamese FCN |
| CAT-NetKwon et al. (2021) | + | DCT | Middle Fusion(Feature Concatenation) | HRNet |
| MVSS-NetChen et al. (2021) | + | Bayar | Late Fusion(Dual Attention) | FCN |
| TruFor Guillaro et al. (2023) | + | Noiseprint | Late Fusion(Feature Concatenation) | Segformer |
| IMDPrompter | + | SRM+Bayar+Noiseprint | Late Fusion(Optimal Prompt Selection +Cross-view Prompt Consistency) | FCN+SAM |

- ManTra-Net Wu et al. (2019): Trained on millions of processed images in a private collection.

- HP-FCN Li & Huang (2019): Trained on a private set of repaired images.

- CR-CNN Yang et al. (2020): Trained on CASIAv2.

- SPAN Hu et al. (2020): Trained using the same data as ManTra-Net and fine-tuned on CASIAv2.

- CAT-Net Kwon et al. (2021): Trained on a joint dataset including CASIAv2, IMD, Fantastic Reality Kniaz et al. (2019), and self-spliced COCO.

For models with publicly available code, we trained them using the code provided by the authors, such as GSR-Net Zhou et al. (2018a). When citing their results, we adopt the original data where appropriate and only use our retrained models when necessary. For models that follow the same evaluation protocol, such as MFCN Salloum et al. (2018) and RGB-N Zhou et al. (2018b), we directly cite results from the same team Yang et al. (2020). For fair comparison, we retrained FCN Long et al. (2015), MVSS-NetDong et al. (2022), and Trufor Guillaro et al. (2023) from scratch on CASIAv2. As previous studies rarely report their image-level performance, these models typically lack an image classification head in their implementations. To obtain a baseline for image-level predictions without modifying their models or code, we adopted the GMP strategy used in MVSS-Net.

## 7.4 SUPPLEMENTARY EXPERIMENTS

**Experimental Setups.** To improve reproducibility, we present the used parameters details in Table 9.

**Overall Performance in Detection and Localization.** Table 10 and Figure 6 shows the overall performance of pixel-level and image-level manipulation detection. We use the harmonic mean of image-level detection F1 and pixel-level localization F1, denoted as C-F1, as our overall performance metric. As shown in Table 3, IMDPrompter achieves the best performance in all settings, with a performance improvement of 45.3% over the next best method. Notably, in the experiments on the COVER dataset, we achieved a performance gain of 110.7%.

**Ablation study of the segmenters for the four prompt views:** We selected several segmenters, including FCN, PSPNet, SETR, and Segformer, for ablation experiments. As shown in the table below, the lightweight FCN based on MobileNet is sufficient to effectively guide SAM's automated prompt learning. The introduction of more powerful segmenters like PSPNet, SETR, and Segformer did not result in a significant improvement in detection performance but instead introduced greater computational overhead. Therefore, we chose the lightweight FCN as our segmenter.

**Computational Complexity Analysis:** As shown in Table 15, I-AUC represents Image-level AUC, I-F1 represents Image-level F1, P-F1 represents Pixel-level F1, and C-F1 represents the harmonic mean of I-F1 and P-F1. It is worth noting that the IMDPrompter mentioned in the main text is based on SAM-H, which has a high computational complexity. Therefore, we implemented a lightweight version, IMDPrompter*, based on Mobile SAM. As can be seen from Table 15, IMDPrompter

Table 9: Parameter Settings Details

| Parameter | Value |
|---|---|
| **Warm-up Learning Rate Scheduler** | |
| Type | LinearLR |
| Start Factor | 1.00E-04 |
| By Epoch | TRUE |
| Begin | 0 |
| End | 1 |
| Convert to Iter Based | TRUE |
| **Main Learning Rate Scheduler** | |
| Type | CosineAnnealingLR |
| T_max | max_epochs |
| By Epoch | TRUE |
| Begin | 1 |
| End | max_epochs |
| **Data Preprocessor** | |
| Mean | [123.675, 116.28, 103.53] |
| Standard Deviation | [58.395, 57.12, 57.375] |
| BGR to RGB | TRUE |
| Padding Value | 0 |
| Segmentation Padding Value | 255 |
| Size | -10,241,024 |
| **Trainer** | |
| Number of Things Classes | 2 |
| Train Batch Size per GPU | 1 |
| Train Number of Workers | 4 |
| Test Batch Size per GPU | 1 |
| Test Number of Workers | 4 |
| Epochs | 100 |

Table 10: Overall Detection and Localization Performance (Com-F1 Score). The best results for each test set are highlighted in bold and second-best values are underlined.

| Method | CASIA | COVER | Columbia | IMD | MEAN |
|---|---|---|---|---|---|
| H-LSTM Bappy et al. (2019) | 0.0 | 0.0 | 3.8 | 0.0 | 0.94 |
| ManTra-Net Wu et al. (2019) | 0.0 | 0.0 | 0.0 | 0.0 | 0.00 |
| CR-CNN Yang et al. (2020) | 30.3 | 18.1 | 41.3 | 23.7 | 28.35 |
| GSR-Net Zhou et al. (2018a) | 0.0 | 0.0 | 4.2 | 0.0 | 1.06 |
| SPAN Hu et al. (2020) | 0.0 | 0.0 | 0.0 | 0.0 | 0.00 |
| CAT-Net Kwon et al. (2021) | 20.0 | 19.7 | 69.1 | 9.5 | 29.59 |
| MVSS-Net Chen et al. (2021) | 56.6 | 31.7 | 71.1 | 30.0 | 47.36 |
| FCN Long et al. (2015) | 53.6 | 18.9 | 30.5 | 23.3 | 31.57 |
| IMDPrompter | **76.8** | **66.8** | **90.3** | **41.3** | **68.82** |
| | +23.2 | +47.9 | +59.8 | +18.0 | +37.25 |

achieved the best performance on almost all metrics, and IMDPrompter* achieved performance second only to IMDPrompter in nearly all metrics.

**Comparison with SAM-based Methods:** As shown in Table 15, We supplement the performance of four SAM-based image segmentation methods: MedSAM, MedSAM-Adapter, AutoSAM, and SAMed. As we can see, these four SAM-based methods exhibit limited performance in the iimage manipulation detection task due to their reliance solely on semantically relevant information. In particular, there is a significant performance gap between them and IMDPrompter, especially on the out-of-domain test sets CVOER, Columbia, and IMD.

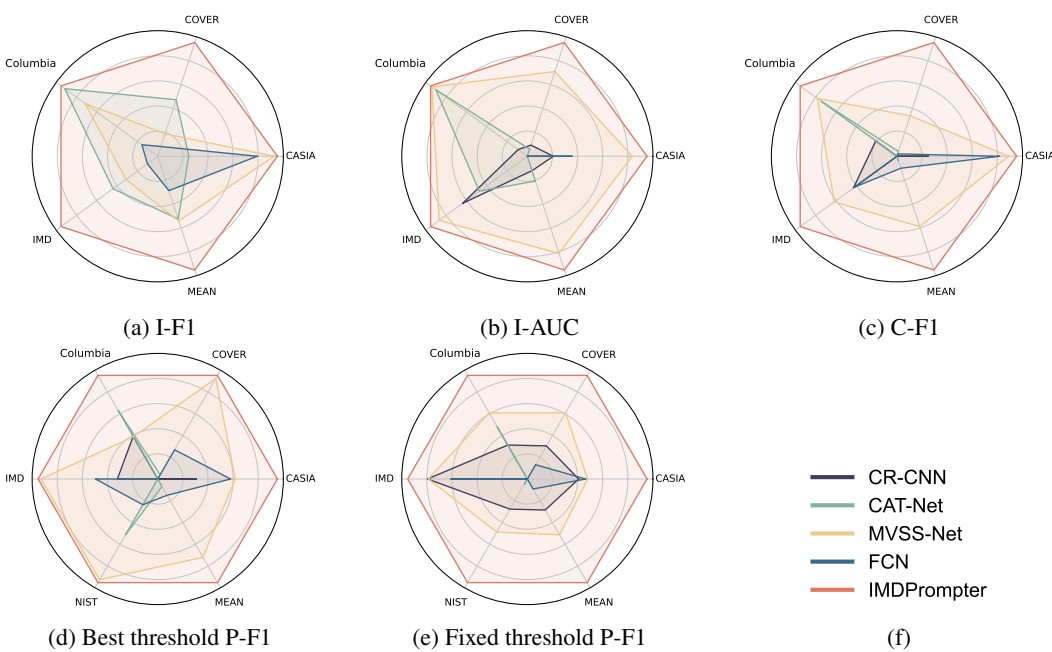

(a) I-F1     (b) I-AUC     (c) C-F1

(d) Best threshold P-F1     (e) Fixed threshold P-F1     (f)

Figure 6: Visualization of Metric Data Comparison between IMDPrompter and State-of-the-Art (SOTA) Methods

Table 11: Impact of JPEG Compression and Gaussian Blur on Pixel-level F1

| Method | JPEG Compression | | | | | | Gaussian Blurs | | | | | |
|---|---|---|---|---|---|---|---|---|---|---|---|---|
| | 100 | 90 | 80 | 70 | 60 | 50 | 0 | 5 | 11 | 17 | 23 | 29 |
| H-LSTM Bappy et al. (2019) | 15.4 | 15.0 | 14.8 | 14.7 | 14.0 | 13.9 | 15.4 | 13.4 | 12.1 | 10.8 | 10.6 | 10.3 |
| ManTra-Net Wu et al. (2019) | 15.5 | 11.9 | 8.2 | 8.0 | 8.7 | 8.6 | 15.5 | 12.0 | 11.3 | 12.0 | 11.6 | 11.2 |
| CR-CNN Yang et al. (2020) | 40.4 | 33.6 | 22.4 | 17.1 | 11.7 | 9.2 | 40.4 | 23.3 | 22.1 | 21.1 | 20.7 | 19.9 |
| GSR-Net Zhou et al. (2018a) | 37.4 | 35.9 | 32.5 | 32.4 | 30.8 | 27.0 | 37.4 | 30.5 | 26.5 | 23.5 | 20.8 | 19.0 |
| SPAN Hu et al. (2020) | 18.2 | 17.1 | 17.1 | 16.6 | 15.7 | 14.9 | 18.2 | 10.7 | 10.6 | 10.0 | 9.6 | 9.6 |
| CAT-Net Kwon et al. (2021) | 13.5 | 2.2 | 2.0 | 1.3 | 0.9 | 0.9 | 13.5 | 9.4 | 9.3 | 6.9 | 6.1 | 6.0 |
| MVSS-Net Chen et al. (2021) | 45.1 | 43.2 | 42.5 | 41.2 | 40.2 | 39.5 | 45.1 | 37.4 | 36.0 | 33.0 | 21.1 | 17.3 |
| IMDPrompter | 76.3 | 73.6 | 72.0 | 71.6 | 70.7 | 70.5 | 76.3 | 71.8 | 71.1 | 68.8 | 56.6 | 51.6 |

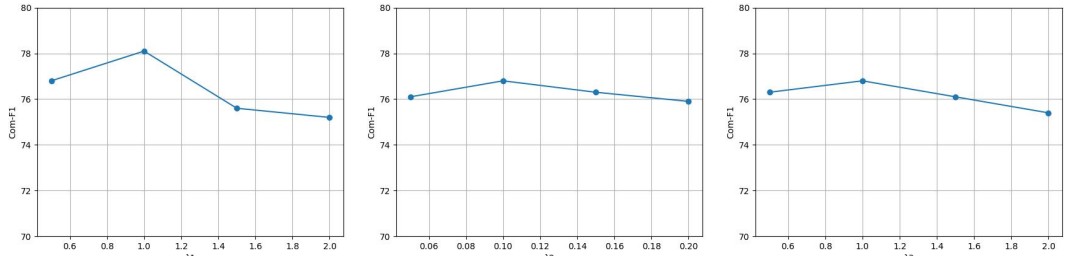

Figure 7: Hyperparametric analysis of $\lambda_1$, $\lambda_2$, and $\lambda_3$

**Proportion of Four Views Selected as Optimal Prompts.** As shown in Table 17, we summarize the proportion of each prompt view being selected as the optimal prompt during training. We observe that the RGB view is selected as the optimal prompt in only 13.9% of cases, indicating that semantically relevant features alone are insufficient to guide SAM's prompt learning. On the other hand, the Noiseprint view is chosen as the optimal prompt in 46.3% of cases, demonstrating that semantically irrelevant information provided by Noiseprint is the most crucial.

Table 12: Impact of JPEG Compression and Gaussian Blur on Image-level F1

| Method | JPEG Compression | | | | | | Gaussian Blur | | | | | |
|---|---|---|---|---|---|---|---|---|---|---|---|---|
| | 100 | 90 | 80 | 70 | 60 | 50 | 0 | 5 | 11 | 17 | 23 | 29 |
| H-LSTM Bappy et al. (2019) | 0.0 | 0.0 | 0.0 | 0.0 | 0.0 | 0.0 | 0.0 | 0.0 | 0.0 | 0.0 | 0.0 | 0.0 |
| ManTra-Net Wu et al. (2019) | 0.0 | 0.0 | 0.0 | 0.0 | 0.0 | 0.0 | 0.0 | 0.0 | 0.0 | 0.0 | 0.0 | 0.0 |
| CR-CNN Yang et al. (2020) | 24.3 | 25.8 | 24.0 | 22.4 | 20.0 | 18.4 | 24.3 | 14.3 | 13.9 | 13.7 | 12.4 | 11.4 |
| GSR-Net Zhou et al. (2018a) | 0.0 | 0.0 | 0.0 | 0.0 | 0.0 | 0.0 | 0.0 | 0.0 | 0.0 | 0.0 | 0.0 | 0.0 |
| SPAN Hu et al. (2020) | 0.0 | 0.0 | 0.0 | 0.0 | 0.0 | 0.0 | 0.0 | 0.0 | 0.0 | 0.0 | 0.0 | 0.0 |
| CAT-Net Kwon et al. (2021) | 38.0 | 17.9 | 17.2 | 15.0 | 12.4 | 11.0 | 38.0 | 11.9 | 11.0 | 10.5 | 10.3 | 10.7 |
| MVSS-Net Chen et al. (2021) | 75.9 | 76.2 | 75.7 | 75.3 | 74.8 | 73.0 | 75.9 | 73.8 | 72.8 | 70.7 | 65.8 | 63.4 |
| IMDPrompter | 77.3 | 77.8 | 77.2 | 76.8 | 76.1 | 74.6 | 77.3 | 75.9 | 74.6 | 73.1 | 69.3 | 66.6 |

Table 13: Impact of JPEG Compression and Gaussian Blur on Com-F1

| Method | JPEG Compression | | | | | | Gaussian Blur | | | | | |
|---|---|---|---|---|---|---|---|---|---|---|---|---|
| | 100 | 90 | 80 | 70 | 60 | 50 | 0 | 5 | 11 | 17 | 23 | 29 |
| H-LSTM Bappy et al. (2019) | 0.0 | 0.0 | 0.0 | 0.0 | 0.0 | 0.0 | 0.0 | 0.0 | 0.0 | 0.0 | 0.0 | 0.0 |
| ManTra-Net Wu et al. (2019) | 0.0 | 0.0 | 0.0 | 0.0 | 0.0 | 0.0 | 0.0 | 0.0 | 0.0 | 0.0 | 0.0 | 0.0 |
| CR-CNN Yang et al. (2020) | 30.3 | 29.2 | 23.2 | 19.4 | 14.8 | 12.3 | 30.3 | 17.7 | 17.0 | 16.6 | 15.5 | 14.5 |
| GSR-Net Zhou et al. (2018a) | 0.0 | 0.0 | 0.0 | 0.0 | 0.0 | 0.0 | 0.0 | 0.0 | 0.0 | 0.0 | 0.0 | 0.0 |
| SPAN Hu et al. (2020) | 0.0 | 0.0 | 0.0 | 0.0 | 0.0 | 0.0 | 0.0 | 0.0 | 0.0 | 0.0 | 0.0 | 0.0 |
| CAT-Net Kwon et al. (2021) | 19.9 | 3.9 | 3.6 | 2.3 | 1.7 | 1.7 | 19.9 | 10.5 | 10.1 | 8.3 | 7.7 | 7.7 |
| MVSS-Net Chen et al. (2021) | 56.6 | 55.1 | 54.4 | 53.2 | 52.3 | 51.2 | 56.6 | 49.7 | 48.2 | 45.0 | 32.0 | 27.2 |
| IMDPrompter | 76.8 | 75.6 | 74.5 | 74.1 | 73.3 | 72.5 | 76.8 | 73.8 | 72.8 | 70.9 | 62.3 | 58.1 |

Table 14: Ablation Study of the Segmenter

| | Params/M | GFLOPs | CASIA | | | | COVER | | | | Columbia | | | | IMD | | | |
|---|---|---|---|---|---|---|---|---|---|---|---|---|---|---|---|---|---|---|
| | | | I-AUC | I-F1 | P-F1 | C-F1 | I-AUC | I-F1 | P-F1 | C-F1 | I-AUC | I-F1 | P-F1 | C-F1 | I-AUC | I-F1 | P-F1 | C-F1 |
| PSPNetZhao et al. (2017) | 365.3 | 1610.3 | 0.979 | 77.4 | 76.5 | 76.9 | 0.794 | 70.3 | 63.7 | 66.8 | 0.984 | 93.7 | 87.8 | 90.7 | 0.673 | 64.1 | 30.6 | 41.4 |
| SETRZheng et al. (2021) | 1591.6 | 2131.6 | 0.985 | 78.1 | 77.1 | 77.6 | 0.801 | 70.9 | 64.3 | 67.4 | 0.988 | 94.1 | 88.2 | 91.1 | 0.679 | 64.5 | 31.1 | 42.0 |
| SegformerXie et al. (2021) | 536.3 | 1769.3 | 0.988 | 78.9 | 77.8 | 78.3 | 0.811 | 71.2 | 65.1 | 68.0 | 0.991 | 94.9 | 88.7 | 91.7 | 0.684 | 65.1 | 32.1 | 43.0 |
| FCNLong et al. (2015) | 347.6 | 1533.2 | 0.978 | 77.4 | 76.4 | 76.9 | 0.795 | 70.2 | 63.7 | 66.8 | 0.984 | 93.6 | 87.6 | 90.5 | 0.670 | 63.8 | 30.6 | 41.4 |

Table 15: Computational Complexity Overhead

| | Params/M | GFLOPs | CASIA | | | | COVER | | | | Columbia | | | | IMD | | | |
|---|---|---|---|---|---|---|---|---|---|---|---|---|---|---|---|---|---|---|
| | | | I-AUC | I-F1 | P-F1 | C-F1 | I-AUC | I-F1 | P-F1 | C-F1 | I-AUC | I-F1 | P-F1 | C-F1 | I-AUC | I-F1 | P-F1 | C-F1 |
| ManTra-NetWu et al. (2019) | 4.0 | 1009.7 | 0.500 | 0.0 | 15.5 | 0.0 | 0.500 | 0.0 | 28.6 | 0.0 | 0.701 | 0.0 | 36.4 | 0.0 | 0.500 | 0.0 | 18.7 | 0.0 |
| SPANHu et al. (2020) | 15.4 | 30.9 | 0.500 | 0.0 | 18.4 | 0.0 | 0.500 | 0.0 | 17.2 | 0.0 | 0.500 | 0.0 | 48.7 | 0.0 | 0.500 | 0.0 | 17.0 | 0.0 |
| MVSS-NetChen et al. (2021) | 146.9 | 160.0 | 0.937 | 75.8 | 45.2 | 56.6 | 0.731 | 24.4 | 45.3 | 31.7 | 0.980 | 80.2 | 63.8 | 71.1 | 0.656 | 35.5 | 26.0 | 30.0 |
| TruforGuillaro et al. (2023) | 67.8 | 90.1 | 0.916 | - | 44.1 | - | 0.770 | - | 19.9 | - | 0.996 | - | 22.3 | - | - | - | 21.0 | - |
| IMDPrompter | 347.6 | 1533.2 | 0.978 | 77.3 | 76.3 | 76.8 | 0.796 | 70.3 | 63.6 | 66.8 | 0.983 | 93.6 | 87.3 | 90.3 | 0.671 | 63.7 | 30.6 | 41.3 |
| IMDPrompter* | 85.8 | 151.3 | 0.951 | 76.1 | 70.3 | 73.1 | 0.779 | 64.5 | 57.7 | 60.9 | 0.980 | 88.3 | 81.6 | 84.8 | 0.667 | 56.9 | 26.9 | 36.5 |

Table 16: Comparison with SAM-based methods.

| | CASIA | | | | COVER | | | | Columbia | | | | IMD | | | |
|---|---|---|---|---|---|---|---|---|---|---|---|---|---|---|---|---|
| | I-AUC | I-F1 | P-F1 | C-F1 | I-AUC | I-F1 | P-F1 | C-F1 | I-AUC | I-F1 | P-F1 | C-F1 | I-AUC | I-F1 | P-F1 | C-F1 |
| MedSAMMa et al. (2024) | 0.861 | 70.3 | 56.4 | 62.6 | 0.563 | 26.3 | 22.1 | 24.0 | 0.791 | 50.1 | 24.6 | 33.0 | 0.513 | 26.1 | 22.4 | 24.1 |
| MedSAM-AdapterWu et al. (2023) | 0.877 | 71.0 | 57.3 | 63.4 | 0.543 | 21.6 | 20.3 | 20.9 | 0.742 | 46.6 | 22.1 | 30.0 | 0.494 | 25.1 | 21.6 | 23.2 |
| AutoSAMShaharabany et al. (2023) | 0.843 | 69.1 | 55.1 | 61.3 | 0.513 | 18.9 | 18.6 | 18.7 | 0.772 | 49.1 | 28.4 | 36.0 | 0.523 | 27.9 | 22.9 | 25.2 |
| SAMedZhang & Liu (2023) | 0.843 | 69.9 | 52.3 | 59.8 | 0.571 | 27.6 | 24.6 | 26.0 | 0.801 | 53.1 | 31.7 | 39.7 | 0.502 | 27.6 | 23.1 | 25.2 |
| IMDPrompter | 0.978 | 77.3 | 76.3 | 76.8 | 0.796 | 70.3 | 63.6 | 66.8 | 0.983 | 93.6 | 87.3 | 90.3 | 0.671 | 63.7 | 30.6 | 41.3 |
| IMDPrompter* | 0.951 | 76.1 | 70.3 | 73.1 | 0.779 | 64.5 | 57.7 | 60.9 | 0.980 | 88.3 | 81.6 | 84.8 | 0.667 | 56.9 | 26.9 | 36.5 |

Table 17: The proportion of each view selected as the optimal view

| Prompt View | RGB | SRM | Bayer | Noiseprint |
|---|---|---|---|---|
| Percent | 13.9% | 18.4% | 21.4% | 46.3% |

**Hyperparametric Analysis** As shown in Figure 7, we conducted hyperparameter analysis on $\lambda_1, \lambda_2$, and $\lambda_3$, ultimately selecting the optimal parameter configuration: $\lambda_1 = 1.0, \lambda_2 = 0.1, \lambda_3 = 1.0$.

Table 18: Performance of Generative Image Manipulation Detection

| Method | CocoGlide | | | |
|---|---|---|---|---|
| | P-F1(best) | P-F1(fixed) | I-AUC | I-Acc |
| ManTraNetWu et al. (2019) | 0.673 | 0.516 | 0.778 | 0.500 |
| SPANHu et al. (2020) | 0.35 | 0.298 | 0.475 | 0.491 |
| MVSS-NetChen et al. (2021) | 0.642 | 0.486 | 0.654 | 0.536 |
| TruforGuillaro et al. (2023) | 0.72 | 0.523 | 0.752 | 0.639 |
| IMDPrompter | 0.746 | 0.539 | 0.781 | 0.652 |

**Performance of Generative Image Manipulation Detection:** Following the work of Trufor, we supplement the performance of IMDPrompter on CocoGlide (image tampering based on diffusion models). As shown in the table below, our IMDPrompter also achieves objective results for AIGC-edited images.

**Impact of Quality Degradation.** As shown in Table 11,12,13, following Dong et al. (2022), we evaluated the robustness of the models under two common image processing operations encountered during the dissemination of images on the internet, namely JPEG compression and Gaussian blur. Comparing these two operations, Gaussian blur has a more significant impact on detection performance, especially when using larger 17x17 scale convolution kernels. Compared to previous methods, our IMDPrompter exhibits better robustness.

**More Visualization.** As shown in Figure 10, we have included additional visualizations.

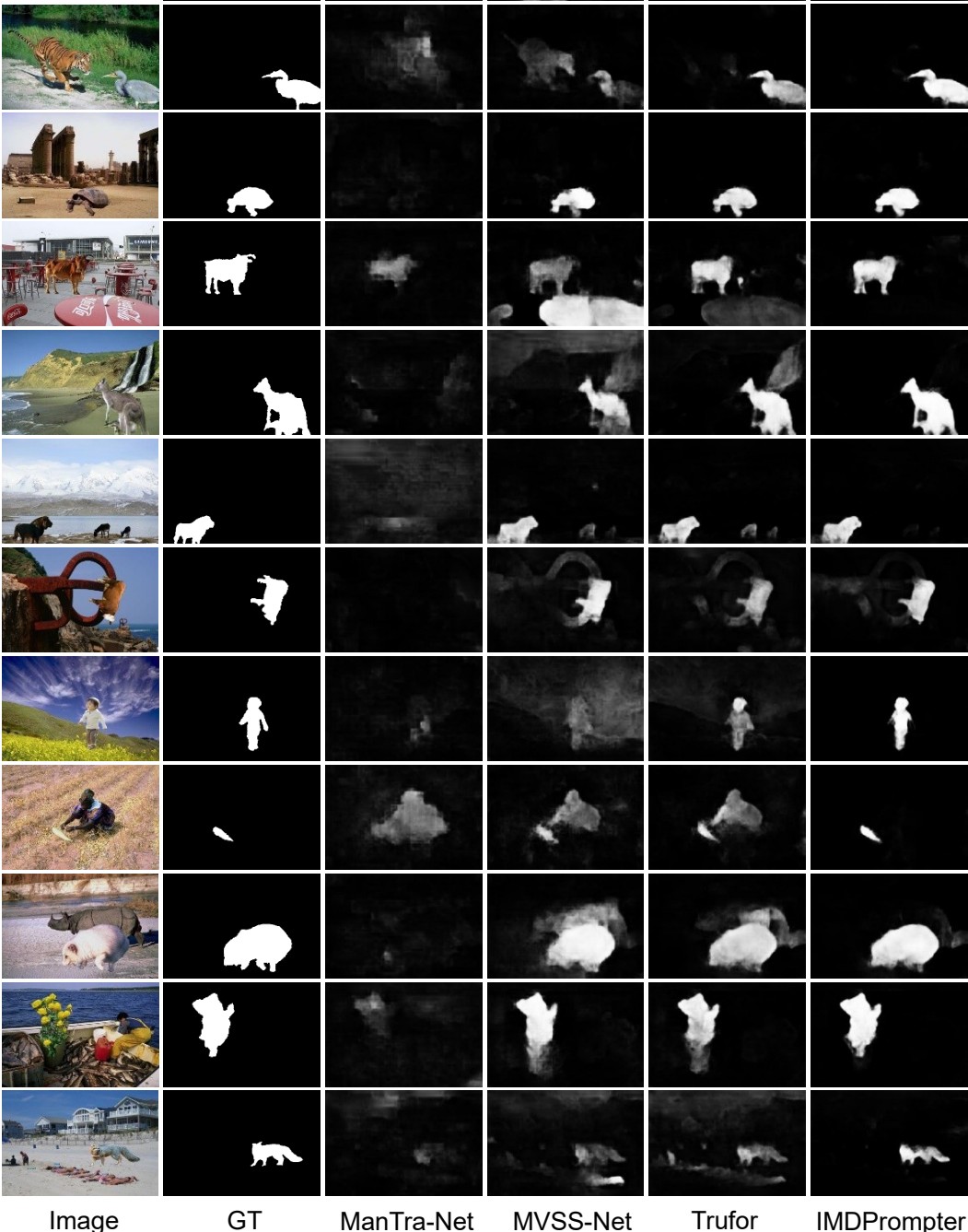

Figure 8: Some qualitative results, compared with the state-of-the-art

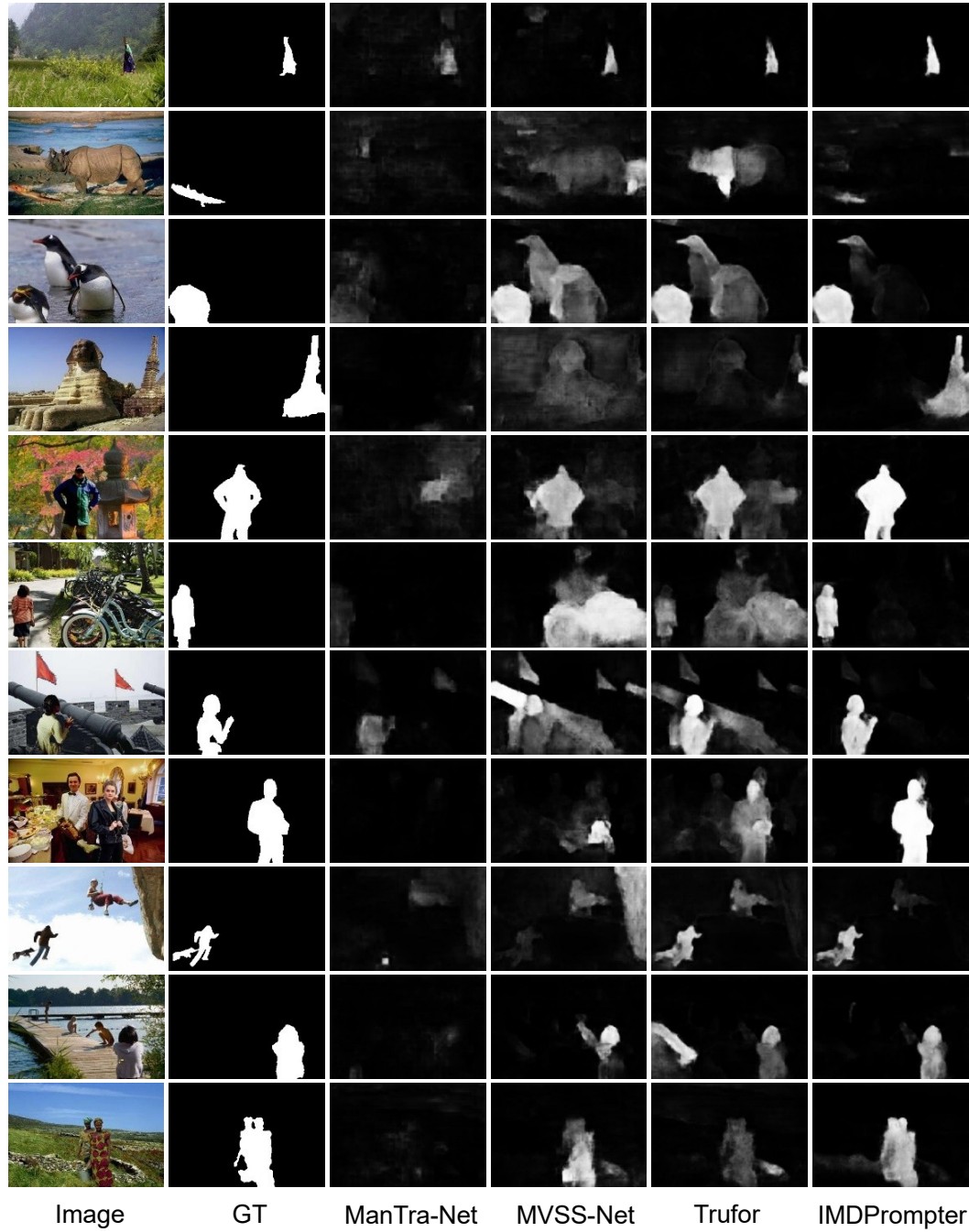

Figure 9: Some qualitative results, compared with the state-of-the-art

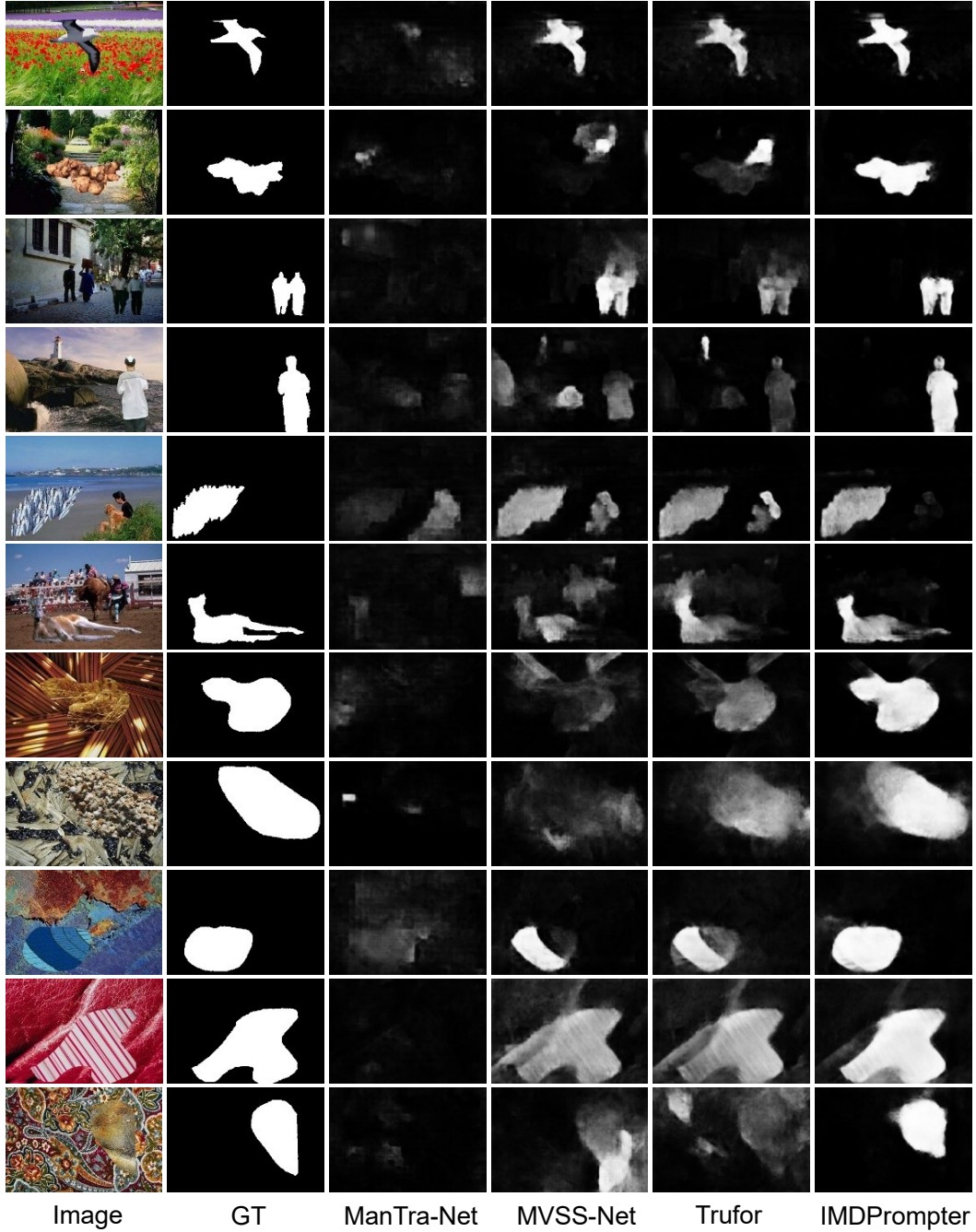

Figure 10: Some qualitative results, compared with the state-of-the-art

