# OpenReview forum: "IMDPrompter: Adapting SAM to Image Manipulation Detection by Cross-View Automated Prompt Learning"
_ICLR.cc/2025/Conference — ICLR 2025 Poster_

### Official Review · Reviewer_tXob · 2024-11-01

**Soundness:** 2
**Presentation:** 3
**Contribution:** 2
**Rating:** 5
**Confidence:** 5

**Summary:**

The paper introduces IMDPrompter, a novel approach for image manipulation detection (IMD) that leverages the Segment Anything Model (SAM). It addresses the challenges of manual prompt reliance and cross-dataset generalization by proposing a cross-view automated prompt learning paradigm. This includes components like Cross-view Feature Perception, Optimal Prompt Selection, and Cross-View Prompt Consistency, which enhance SAM's ability to generate accurate masks for detection and localization without manual guidance. The method is validated across five datasets, demonstrating its effectiveness in both in-distribution and out-of-distribution image manipulation detection and localization.

**Strengths:**

The primary strength of this paper lies in its application of the Segment Anything Model (SAM) to the field of image manipulation detection (IMD), introducing a cross-view automatic prompt learning paradigm. This paradigm significantly enhances the automation and generalization across datasets in IMD tasks through components such as automated prompt generation, optimal prompt selection, and cross-view prompt consistency. Furthermore, the paper demonstrates the effectiveness of the proposed method through extensive experiments across five different datasets, testing not only the model's in-distribution performance but also its robustness in out-of-distribution scenarios. These contributions not only advance the technology of image manipulation detection but also provide a powerful new tool for the field of multimedia forensics.

**Weaknesses:**

1. The technical details about the Cross-View Consistency Enhancement Module (CPC) implementation are insufficient. The paper does not clearly explain how different consistency loss formulations impact the results, and there is no visualization or analysis demonstrating how cross-view consistency is maintained throughout the detection process.
2. The comparison baselines are relatively outdated, missing important recent work from 2024 and other foundation model-based approaches. This limits the comprehensiveness of the comparative analysis and makes it difficult to assess the method's standing against the latest advancements in the field.
3. The paper shows limited discussion of model robustness to distribution shifts and lacks experiments demonstrating how the model adapts to real-world scenarios where data distribution varies. There is no clear mechanism described for handling domain adaptation.
4. The multi-view fusion analysis is incomplete. The paper lacks detailed ablation studies that quantify the individual contribution of each view, and there is no discussion of the computational trade-offs associated with different view combinations.
5. The paper lacks justification for selecting SRM, Bayer, and Noiseprint as noise perspectives. There is no analysis of their specific effectiveness for different types of image manipulations, nor any comparison with other potential noise perspectives that could potentially achieve similar or better results.

**Questions:**

1. Can you provide more detailed technical information about the consistency loss formulation in CPC? It would be helpful to see visual examples of how CPC affects the prompt generation process and its impact on the final detection results.
2. Why were certain recent methods excluded from the comparison? Have you considered comparing with methods published in 2024 or other SAM-based approaches?
3. How does IMDPrompter handle images from different domains in practical applications? What mechanisms could be added to make the model more adaptive to distribution shifts?
4. Can you provide comprehensive ablation studies showing the contribution of each view and how different view combinations affect the model's performance? This should include an analysis of the computational overhead associated with each additional view.
5. Could you explain the rationale behind choosing SRM, Bayer, and Noiseprint specifically as noise perspectives? Have experiments been conducted with other noise perspectives, and if so, what were the results?

---

> ### Author Response · Authors · 2024-11-24
>
> **Weakness 1: The technical details about the Cross-View Consistency Enhancement Module (CPC) implementation are insufficient. The paper does not clearly explain how different consistency loss formulations impact the results, and there is no visualization or analysis demonstrating how cross-view consistency is maintained throughout the detection process.**
>
> Thank you for your constructive suggestions. We have supplemented explanations from the following four aspects:
>
> - **Technical Details of the CPC Loss Function:** Referring to lines 303 and 304 of the original paper, we selected **Focal Loss** as the consistency constraint loss.
>
>   The detailed mathematical expression for **Focal Loss** is as follows:
>
>   \[
>   FL(p_t) = - \alpha_t (1 - p_t)^\gamma \log(p_t)
>   \]
>
>   Where:
>   - \( p_t \) is defined as:
>     \[
>     p_t =
>     \begin{cases}
>     p & \text{if the true label } y = 1, \\
>     1 - p & \text{if the true label } y = 0.
>     \end{cases}
>     \]
>     Here, \( p \) is the predicted probability for the positive class (\( y=1 \)).
>
>   - \( \alpha_t \): A weighting factor for class imbalance (\( \alpha_t \in [0, 1] \)).
>   - \( \gamma \): The focusing parameter, controlling the reduction of loss contribution for well-classified examples (\( \gamma \geq 0 \)).
>
>   This detailed formulation makes Focal Loss highly effective for imbalanced datasets or cases where misclassified samples are more critical.
>
> - **Impact of Different CPC Implementations:** As shown in the table below, we compared MSE Loss, L1 Loss, Smooth L1 Loss, and KL Divergence Loss and Focal loss. Ultimately, Focal Loss achieved the best performance; therefore, we selected Focal Loss as our consistency constraint loss function.
>
> |  |   |   | CASIA |       |        | COVER |       |       |        | Columbia |       |       |        |  IMD  |       |       |
> |:-------------------------:|:------:|:-----:|:-----:|:-----:|:------:|:-----:|:-----:|:-----:|:------:|:--------:|:-----:|:-----:|:------:|:-----:|:-----:|:-----:|
> |          MSE          | 0.961  | 76.0  | 75.0  | 75.5  | 0.782  | 69.1  | 62.5  | 65.7  | 0.966  |   92.0   | 85.8  | 88.8  | 0.660  | 62.6  | 30.1  | 40.6  |
> |    L1   | 0.962  | 76.1  | 75.1  | 75.6  | 0.783  | 69.2  | 62.6  | 65.7  | 0.967  |   92.1   | 85.9  | 88.9  | 0.660  | 62.7  | 30.1  | 40.6  |
> |   Smooth L1     | 0.965  | 76.3  | 75.3  | 75.8  | 0.786  | 69.4  | 62.8  | 65.9  | 0.970  |   92.4   | 86.2  | 89.1  | 0.662  | 62.9  | 30.2  | 40.8  |
> |     KL divergence   | 0.967  | 76.4  | 75.5  | 76.0  | 0.787  | 69.5  | 62.9  | 66.1  | 0.972  |   92.6   | 86.3  | 89.3  | 0.664  | 63.0  | 30.3  | 40.8  |
> |     Cross-Entropy   | 0.968  | 76.5  | 75.5  | 76.0  | 0.788  | 69.6  | 63.0  | 66.1  | 0.973  |   92.7   | 86.4  | 89.4  | 0.664  | 63.1  | 30.3  | 40.9  |
> |         Focal loss        | 0.978  | 77.3  | 76.3  | 76.8  | 0.796  | 70.3  | 63.6  | 66.8  | 0.983  |   93.6   | 87.3  | 90.3  | 0.671  | 63.7  | 30.6  | 41.3  |
>
>   To further analyze how CPC affects the prompt generation process and the final detection results, we constructed a baseline method based on weighted averaging of multi-view outputs, then sequentially introduced OPS and CPC to explore how CPC influences the prompt generation process and the final detection results.
>
> - **Cosine Similarity Relative to the Optimal Prompt:** After using OPS, the cosine similarity between prompts of each view and the optimal prompt did not increase significantly. This indicates that using only OPS is insufficient to align the prompts with the optimal view. However, when both OPS and CPC are used together, the cosine similarity between each view and the optimal view increases significantly (all exceeding 0.95), indicating that using both OPS and CPC effectively aligns the prompts of each view with the optimal view.
>
>   |                | **RGB** | **SRM** | **Bayar** | **Noiseprint** |
>   |----------------|---------|---------|-----------|----------------|
>   | **Weighted Average** | 0.82    | 0.85    | 0.83      | 0.92           |
>   | **OPS**         | 0.81    | 0.86    | 0.83      | 0.99           |
>   | **OPS+CPC**     | 0.95    | 0.97    | 0.96      | 0.99           |
>
> - **F1 Metrics Relative to GT:** Furthermore, we evaluated the F1 metrics of the prompt masks of each view against the ground truth (GT). Compared to the weighted average strategy, OPS prevents the optimal prompt from being degraded by inaccurate prompts, thereby improving the performance of final image tampering detection. At the same time, we found that when only OPS is introduced, the F1 metrics of the prompt masks for each view do not improve, indicating that using only OPS cannot achieve cross-view consistency enhancement. When both OPS and CPC are used, not only are the final image tampering detection metrics improved, but the prompt F1 of each view is also enhanced. This indicates that using both OPS and CPC achieves cross-view consistency enhancement, bringing each view closer to the optimal view.

---

> ### Author Response · Authors · 2024-11-24
>
> |                | **RGB** | **SRM** | **Bayar** | **Noiseprint** | **SAM Output** |
>   |----------------|---------|---------|-----------|----------------|----------------|
>   | **Weighted Average** | 42.9    | 58.1    | 54.3      | 61.7           | 73.8           |
>   | **OPS**         | 42.5    | 58.3    | 55.0      | 62.4           | 75.4           |
>   | **OPS+CPC**     | 65.7    | 66.1    | 65.4      | 67.9           | 76.8           |
>
> ---
>
> **Weakness 2: The comparison baselines are relatively outdated, missing important recent work from 2024 and other foundation model-based approaches. This limits the comprehensiveness of the comparative analysis and makes it difficult to assess the method's standing against the latest advancements in the field.**
>
> Thank you for your constructive suggestions. We have supplemented explanations from the following two aspects:
>
> - **Comparison with Important Recent Works from 2024:** As shown in the table below, we adopted the same combined training set as the latest 2024 method UnionFormer[1](CVPR2024) and evaluated performance on each test set. Our method achieved state-of-the-art (SOTA) performance.
>
>   **Pixel-Level F1 Experimental Analysis:**
>
> |             | Optimal threshold |          |       |       |           |       | Fixed threshold(0.5) |          |       |       |           |       |
> |:-----------:|:-----------------:|:--------:|:-----:|:-----:|:---------:|:-----:|:--------------------:|:--------:|:-----:|:-----:|:---------:|:-----:|
> |             |      Columbia     | Coverage | CASIA |  NIST | CoCoGlide |  AVG  |       Columbia       | Coverage | CASIA |  NIST | CoCoGlide |  AVG  |
> |  CAT-Net v2 |       92.3        |   58.2   | 85.2  | 41.7  |   60.3    | 67.5  |         85.9         |   38.1   | 75.2  | 30.8  |   43.4    | 54.7  |
> |    Trufor   |       91.4        |   73.5   | 82.2  | 47.0  |   72.0    | 73.2  |         85.9         |   60.0   | 73.7  | 39.9  |   52.3    | 62.4  |
> | UnionFormer |       92.5        |   72.0   | 86.3  | 48.9  |   74.2    | 74.8  |         86.1         |   59.2   | 76.0  | 41.3  |   53.6    | 63.2  |
> | IMDPrompter |       92.7        |   74.0   | 87.1  | 50.1  |   74.0    | 75.6  |         86.7         |   60.3   | 76.8  | 42.4  |   52.9    | 63.8  |
>
>   **Image-Level Detection Experimental Analysis:**
>
> |             |          |          | Image-level  AUC |        |           |        |          |          | Accuracy |        |           |        |
> |:-----------:|:--------:|:--------:|:--------------------------:|:------:|:---------:|:------:|:--------:|:--------:|:--------:|:------:|:---------:|:------:|
> |             | Columbia | Coverage |            CASIA           |  NIST  | CoCoGlide |   AVG  | Columbia | Coverage |   CASIA  |  NIST  | CoCoGlide |   AVG  |
> |  CAT-Net v2 |  0.977   |  0.680   |           0.942            | 0.750  |   0.667   | 0.803  |  0.803   |  0.635   |  0.838   | 0.597  |   0.580   | 0.691  |
> |    Trufor   |  0.996   |  0.770   |           0.916            | 0.760  |   0.752   | 0.839  |  0.984   |  0.680   |  0.813   | 0.662  |   0.639   | 0.756  |
> | UnionFormer |  0.998   |  0.783   |           0.951            | 0.793  |   0.797   | 0.864  |  0.979   |  0.694   |  0.843   | 0.680  |   0.682   | 0.776  |
> | IMDPrompter |  0.998   |  0.791   |           0.962            | 0.801  |   0.801   | 0.871  |  0.984   |  0.703   |  0.851   | 0.682  |   0.680   | 0.780  |
>
>   **Pixel-Level AUC Experimental Analysis:**
>
> |             | Columbia | Coverage |  CASIA |  NIST  |   IMD  |   AVG  |
> |:-----------:|:--------:|:--------:|:------:|:------:|:------:|:------:|
> |    Trufor   |  0.947   |  0.925   | 0.957  | 0.877  |    -   | 0.927  |
> | UnionFormer |  0.989   |  0.945   | 0.972  | 0.881  | 0.860  | 0.929  |
> | IMDPrompter |  0.990   |  0.948   | 0.978  | 0.890  | 0.864  | 0.934  |
>
> - **Comparison with Other Foundation Model-Based Methods:** Referring to Supplementary Material Table 16, we have added comparisons with foundation model-based methods such as MedSAM, MedSAM-Adapter, AutoSAM, and SAMed. IMDPrompter outperforms these methods in metrics like I-AUC, I-F1, P-F1, and Com-F1.
>
> [1]  Li, Shuaibo, et al. "UnionFormer: Unified-Learning Transformer with Multi-View Representation for Image Manipulation Detection and Localization." Proceedings of the IEEE/CVF Conference on Computer Vision and Pattern Recognition. 2024.

---

> ### Author Response · Authors · 2024-11-24
>
> **Weakness 3: The paper shows limited discussion of model robustness to distribution shifts and lacks experiments demonstrating how the model adapts to real-world scenarios where data distribution varies. There is no clear mechanism described for handling domain adaptation.**
>
> Thank you for your constructive suggestions. We have provided a detailed analysis from the following three aspects:
>
> - **Robustness Analysis for Gaussian Blur and JPEG Compression:** Referring to Figure 5 in the main paper, we tested the performance of IMDPrompter and other methods like MVSS-Net under varying degrees of JPEG compression and Gaussian Blur. We found that IMDPrompter achieved the best performance across all settings, demonstrating strong robustness to Gaussian Blur and JPEG compression.
>
> - **Robustness Analysis for Lighting and WEBP Compression:** Referring to the table below, we tested the performance of IMDPrompter and other methods like MVSS-Net under different lighting conditions and WEBP compression. We found that IMDPrompter achieved the best performance across all settings, demonstrating strong robustness to various lighting conditions and WEBP compression.
>
>
> |      P-F1     |       |       | Lighting |       |       |       |       |       | WEBP compress  |       |       |       |
> |:-------------:|:-----:|:-----:|:--------:|:-----:|:-----:|:-----:|:-----:|:-----:|:--------------:|:-----:|:-----:|:-----:|
> |     Method    |  -75  |  -50  |    -25   |   +0   |   +25  |   +50  |   100   |   90   |       80       |   70  |   60  |   50  |
> |   H-LSTM[57]  | 14.3  | 14.7  |   15.4   | 15.4  | 15.3  | 15.0  | 15.4  | 13.2  |      12.1      | 10.8  | 10.6  | 10.1  |
> | ManTra-Net[6] | 15.1  | 15.2  |   15.3   | 15.5  | 15.4  | 15.2  | 15.5  | 11.8  |      11.2      | 11.8  | 11.5  | 11.1  |
> |   CR-CNN[17]  | 39.0  | 39.4  |   39.7   | 40.4  | 40.1  | 39.0  | 40.4  | 22.9  |      21.9      | 20.9  | 20.3  | 19.8  |
> |  GSR-Net[54]  | 35.3  | 36.1  |   36.5   | 37.4  | 37.3  | 37.0  | 37.4  | 30.4  |      26.3      | 23.3  | 20.8  | 18.9  |
> |    SPAN[16]   | 17.4  | 17.4  |   17.7   | 18.2  | 18.1  | 17.7  | 18.2  | 10.5  |      10.4      | 10.0  |  9.4  |  9.5  |
> |  CAT-Net[58]  | 12.5  | 12.8  |   13.2   | 13.5  | 13.2  | 12.8  | 13.5  |  9.3  |      9.2       |  6.9  |  6.0  |  5.9  |
> |  MVSS-Net[59] | 43.3  | 44.3  |   45.0   | 45.1  | 44.6  | 44.2  | 45.1  | 37.0  |      35.9      | 32.9  | 21.1  | 17.2  |
> |  IMDPrompter  | 74.3  | 74.8  |   76.0   | 76.3  | 75.5  | 74.8  | 76.3  | 71.2  |      70.2      | 67.5  | 55.7  | 50.9  |
>
> |      I-F1     |       |       | Lighting |       |       |       |       |       | WEBP compress |       |       |       |
> |:-------------:|:-----:|:-----:|:--------:|:-----:|:-----:|:-----:|:-----:|:-----:|:-------------:|:-----:|:-----:|:-----:|
> |     Method    |  -75  |  -50  |    -25   |   +0   |   +25  |   +50  |   100   |   90   |       80       |   70  |   60  |   50  |
> |   H-LSTM[57]  |  0.0  |  0.0  |   0.0    |  0.0  |  0.0  |  0.0  |  0.0  |  0.0  |      0.0      |  0.0  |  0.0  |  0.0  |
> | ManTra-Net[6] |  0.0  |  0.0  |   0.0    |  0.0  |  0.0  |  0.0  |  0.0  |  0.0  |      0.0      |  0.0  |  0.0  |  0.0  |
> |   CR-CNN[17]  | 23.3  | 23.7  |   24.3   | 24.3  | 24.1  | 23.5  | 24.3  | 14.5  |     14.1      | 13.9  | 12.4  | 11.4  |
> |  GSR-Net[54]  |  0.0  |  0.0  |   0.0    |  0.0  |  0.0  |  0.0  |  0.0  |  0.0  |      0.0      |  0.0  |  0.0  |  0.0  |
> |    SPAN[16]   |  0.0  |  0.0  |   0.0    |  0.0  |  0.0  |  0.0  |  0.0  |  0.0  |      0.0      |  0.0  |  0.0  |  0.0  |
> |  CAT-Net[58]  | 37.0  | 37.5  |   37.8   | 38.0  | 37.2  | 36.4  | 38.0  | 12.1  |     11.0      | 10.6  | 10.4  | 10.7  |
> |  MVSS-Net[59] | 72.4  | 74.3  |   74.8   | 75.9  | 73.7  | 72.7  | 75.9  | 74.1  |     73.1      | 70.9  | 66.6  | 64.0  |
> |  IMDPrompter  | 74.7  | 74.9  |   76.8   | 77.3  | 75.9  | 74.0  | 77.3  | 76.8  |     75.3      | 74.5  | 70.1  | 67.5  |

---

> ### Author Response · Authors · 2024-11-24
>
> **Com-F1 Metrics for Lighting and WEBP Compression**
>
> |     Com-F1    |       |       | Lighting |       |       |       |       |       | WEBP compress |       |       |       |
> |:-------------:|:-----:|:-----:|:--------:|:-----:|:-----:|:-----:|:-----:|:-----:|:-------------:|:-----:|:-----:|:-----:|
> |     Method    |  -75  |  -50  |    -25   |   +0   |   +25  |   +50  |  100   |   90   |       80       |   70  |   60  |   50  |
> |   H-LSTM[57]  |  0.0  |  0.0  |   0.0    |  0.0  |  0.0  |  0.0  |  0.0  |  0.0  |      0.0      |  0.0  |  0.0  |  0.0  |
> | ManTra-Net[6] |  0.0  |  0.0  |   0.0    |  0.0  |  0.0  |  0.0  |  0.0  |  0.0  |      0.0      |  0.0  |  0.0  |  0.0  |
> |   CR-CNN[17]  | 29.3  | 29.6  |   30.1   | 30.3  | 30.1  | 29.4  | 30.3  | 17.8  |     17.2      | 16.7  | 15.4  | 14.5  |
> |  GSR-Net[54]  |  0.0  |  0.0  |   0.0    |  0.0  |  0.0  |  0.0  |  0.0  |  0.0  |      0.0      |  0.0  |  0.0  |  0.0  |
> |    SPAN[16]   |  0.0  |  0.0  |   0.0    |  0.0  |  0.0  |  0.0  |  0.0  |  0.0  |      0.0      |  0.0  |  0.0  |  0.0  |
> |  CAT-Net[58]  | 18.9  | 19.1  |   19.6   | 19.9  | 19.5  | 19.0  | 19.9  | 10.5  |     10.0      |  8.4  |  7.6  |  7.6  |
> |  MVSS-Net[59] | 54.6  | 55.5  |   56.2   | 56.6  | 55.6  | 55.0  | 56.6  | 49.3  |     48.2      | 44.9  | 32.0  | 27.1  |
> |  IMDPrompter  | 73.8  | 74.8  |   76.4   | 76.8  | 75.7  | 74.4  | 76.8  | 73.9  |     72.7      | 70.8  | 62.1  | 58.1  |
>
> - **Mechanisms for Handling Domain Shifts in IMDPrompter:** Utilizing four mechanisms—large-scale pre-trained SAM's generalization ability, multi-view design, optimal prompt selection, and cross-view consistency constraints—we analyze each mechanism as follows:
>
>   - **Large-Scale Pre-Trained SAM’s Generalization Ability:** SAM's extensive pre-training provides strong generalization across various domains, offering potential priors for image tampering detection and refining detection results. We constructed a baseline method, FCN+, which uses multi-view design but does not utilize SAM. As shown in the table below, FCN+ lacks the extensive pre-training of SAM and thus has weaker generalization to other domains. Consequently, under domain shift conditions, its detection accuracy is significantly lower than that of IMDPrompter.
>
>     | Method          | 100.0 | 90.0 | 80.0 | 70.0 | 60.0 | 50.0 |
>     |-----------------|-------|------|------|------|------|------|
>     | **FCN+**        | 47.2  | 46.3 | 45.3 | 44.4 | 43.5 | 42.7 |
>     | **IMDPrompter** | 76.3  | 73.6 | 72.0 | 71.6 | 70.7 | 70.5 |
>
>   - **Multi-View Design:** Combining information from RGB views, we constructed a baseline method IMDPrompter-single, which uses only the RGB view prompts for SAM. In domain shift scenarios, the detection accuracy of IMDPrompter-single is significantly lower than that of the multi-view IMDPrompter.
>
>     | Method               | 100.0 | 90.0 | 80.0 | 70.0 | 60.0 | 50.0 |
>     |----------------------|-------|------|------|------|------|------|
>     | **IMDPrompter-single** | 66.2  | 64.1 | 62.4 | 61.9 | 60.1 | 71.2 |
>     | **IMDPrompter**        | 76.3  | 73.6 | 72.0 | 71.6 | 70.7 | 70.5 |
>
>   - **Optimal Prompt Selection:** For different domain shifts, the most suitable prompt views vary. To select the most robust prompt for the current domain shift, we implemented an optimal prompt selection process based on minimizing segmentation loss. We found that by introducing the optimal prompt selection module, IMDPrompter achieved better adaptability to domain shifts.
>
>   - **Cross-View Prompt Consistency:** For suboptimal prompts under the current domain shift, cross-view prompt consistency can be used to align them with the optimal prompts, thereby enhancing robustness to domain shifts. The optimal prompt selection module can select the prompts from multiple views that are more adaptable to the current domain shift. Through cross-view consistency constraints, all views achieve strong adaptability to the current domain shift.
>
>     | Method                  | 100.0 | 90.0 | 80.0 | 70.0 | 60.0 | 50.0 |
>     |-------------------------|-------|------|------|------|------|------|
>     | **without CPC and OPS** | 69.7  | 0.0  | 62.4 | 61.9 | 60.1 | 71.2 |
>     | **+OPS**                | 71.3  | 67.6 | 65.1 | 64.3 | 63.7 | 64.2 |
>     | **+OPS and +CPC**       | 76.3  | 73.6 | 72.0 | 71.6 | 70.7 | 70.5 |

---

> ### Author Response · Authors · 2024-11-24
>
> **Weakness 4: The multi-view fusion analysis is incomplete. The paper lacks detailed ablation studies that quantify the individual contribution of each view, and there is no discussion of the computational trade-offs associated with different view combinations.**
>
>    | RGB  | Noiseprint | Bayar | SRM | CASIA | COVER | Columbia | IMD | Training Time (h) | Params (M) |
>    |------|------------|-------|-----|-------|-------|----------|-----|-------------------|------------|
>    | √    |            |       |     | 0.856 | 73.6  | 69.6     | 71.5| 12.31             | 318.10     |
>    |      | √          |       |     | 0.903 | 75.9  | 73.6     | 74.7| 12.31             | 318.10     |
>    |      |            | √     |     | 0.863 | 74.3  | 70.6     | 72.4| 12.31             | 318.10     |
>    |      |            |       | √   | 0.896 | 75.1  | 71.6     | 73.3| 12.31             | 318.10     |
>    | √    | √          |       |     | 0.941 | 76.9  | 75.2     | 76.0| 12.56             | 327.90     |
>    | √    |            | √     |     | 0.936 | 76.1  | 74.6     | 75.3| 12.56             | 327.90     |
>    | √    |            |       | √   | 0.937 | 76.4  | 74.5     | 75.4| 12.56             | 327.90     |
>    | √    | √          | √     |     | 0.962 | 77.1  | 75.9     | 76.5| 12.70             | 337.80     |
>    | √    | √          | √     | √   | 0.978 | 77.3  | 76.3     | 76.8| 12.81             | 347.60     |
>
> - **Ablation Studies on the Contribution of Each View:** Referring to the table above, we have added detailed ablation studies for different view combinations. We found that when only one view is used, using only the Noiseprint view achieves better detection quality. Additionally, when the RGB view is combined with any noise view, IMDPrompter achieves SOTA performance on most metrics. When RGB, Noiseprint, Bayar, and SRM views are used together, IMDPrompter achieves the best performance while maintaining a slight increase in computational overhead.
>
> - **Computational Trade-Offs of Different View Combinations:** Referring to Table 5 in the main paper and the table above, when RGB, Noiseprint, Bayar, and SRM views are used together, IMDPrompter achieves the best performance with only a 0.5-hour increase in training time and a 9.3% increase in the number of parameters, thereby achieving optimal performance.

---

> ### Author Response · Authors · 2024-11-24
>
> **Weakness 5: The paper lacks justification for selecting SRM, Bayer, and Noiseprint as noise perspectives. There is no analysis of their specific effectiveness for different types of image manipulations, nor any comparison with other potential noise perspectives that could potentially achieve similar or better results.**
>
> - **Criteria for Selecting Types of Image Processing Information:** The SRM, Bayar, and Noiseprint views aim to introduce semantic-agnostic features from multiple perspectives. The definition of semantic-agnostic features is as follows: "Low-level artifacts are caused by the in-camera acquisition process, such as the sensor, the lens, the color filter array, or the JPEG quantization tables. In the Image Manipulation Detection task, semantic-agnostic features refer to features that prominently represent low-level artifact information. These features are unrelated to the semantic content of the image. For tampered and untampered regions of an image, there is a significant difference in feature distribution. Common methods for extracting such features include SRM, Bayar, DCT, and Noiseprint." Therefore, we can conclude that views which better highlight low-level artifact information, thereby exhibiting stronger differences in feature distribution between tampered and untampered regions of an image, are the views we seek.
>
> - **Reasons for Choosing SRM, Bayar, Noiseprint, and RGB as Prompt Views:** The selection of semantic-agnostic features needs to meet the following criteria: (1) Each position in the noise view corresponds one-to-one with the corresponding position in the original image, and (2) the ability to prominently display low-level artifact information so that there is a stronger difference in feature distribution between tampered and untampered regions of an image. Taking these two conditions into account, we selected five semantic-agnostic views: Noiseprint, SRM, Bayar, DCT, and Wavelet. The ablation experiments under various feature combinations are shown in the table below. When each view is used individually, we found that using only the Wavelet view resulted in the worst performance on in-domain test sets, and its performance on out-of-domain test sets was only slightly better than using the RGB view alone. This indicates that the semantic-agnostic features extracted by the Wavelet view provide limited useful information for our image tampering detection task. When we sequentially introduced the Noiseprint, Bayar, and SRM views based on the RGB view, the metrics on each dataset gradually improved. However, when the DCT view was introduced, performance did not increase significantly. Further introducing the Wavelet view did not lead to performance improvements on any dataset. Considering computational overhead and detection accuracy, we selected RGB along with Noiseprint, SRM, and Bayar as our prompt views.

---

> ### Author Response · Authors · 2024-11-24
>
> | RGB | Noiseprint | Bayar | SRM | DCT | Wavelet |  CASIA |       |       |       |  COVER |       |       |       | Columbia |       |       |       |   IMD  |       |       |       | training time/h | Params/M |
> |:---:|:----------:|:-----:|:---:|:---:|:-------:|:------:|:-----:|:-----:|:-----:|:------:|:-----:|:-----:|:-----:|:--------:|:-----:|:-----:|:-----:|:------:|:-----:|:-----:|:-----:|:---------------:|:--------:|
> |     |            |       |     |     |         |  I-AUC |  I-F1 |  P-F1 |  C-F1 |  I-AUC |  I-F1 |  P-F1 |  C-F1 |   I-AUC  |  I-F1 |  P-F1 |  C-F1 |  I-AUC |  I-F1 |  P-F1 |  C-F1 |                 |          |
> |  √  |            |       |     |     |         | 0.856  | 73.6  | 69.6  | 71.5  | 0.556  | 19.6  | 38.6  | 26.0  |  0.786   | 53.6  | 24.9  | 34.0  | 0.531  | 28.1  | 21.2  | 24.2  |      12.31      |  318.1   |
> |     |      √     |       |     |     |         | 0.903  | 75.9  | 73.6  | 74.7  | 0.761  | 62.9  | 58.7  | 60.7  |  0.946   | 88.6  | 83.5  | 86.0  | 0.636  | 59.9  | 27.9  | 38.1  |      12.31      |  318.1   |
> |     |            |   √   |     |     |         | 0.863  | 74.3  | 70.6  | 72.4  | 0.706  | 61.3  | 56.3  | 58.7  |  0.889   | 83.4  | 81.6  | 82.5  | 0.601  | 54.6  | 25.7  | 34.9  |      12.31      |  318.1   |
> |     |            |       |  √  |     |         | 0.896  | 75.1  | 71.6  | 73.3  | 0.713  | 59.6  | 55.6  | 57.5  |  0.906   | 84.6  | 79.3  | 81.9  | 0.593  | 56.3  | 24.3  | 33.9  |      12.31      |  318.1   |
> |     |            |       |     |  √  |         | 0.861  | 74.1  | 70.7  | 72.4  | 0.613  | 32.4  | 47.3  | 38.5  |  0.843   | 75.6  | 70.7  | 73.1  | 0.553  | 36.9  | 22.1  | 27.6  |      12.31      |  318.1   |
> |     |            |       |     |     |    √    | 0.746  | 66.7  | 62.1  | 64.3  | 0.576  | 23.6  | 41.3  | 30.0  |  0.791   | 54.9  | 26.4  | 35.7  | 0.560  | 30.9  | 23.6  | 26.8  |      12.31      |  318.1   |
> |  √  |      √     |       |     |     |         | 0.941  | 76.9  | 75.2  | 76.0  | 0.781  | 66.8  | 61.9  | 64.3  |  0.969   | 91.7  | 85.4  | 88.4  | 0.655  | 61.5  | 28.8  | 39.2  |      12.56      |  327.9   |
> |  √  |      √     |   √   |     |     |         | 0.962  | 77.1  | 75.9  | 76.5  | 0.788  | 68.7  | 62.4  | 65.4  |  0.977   | 92.4  | 86.5  | 89.4  | 0.660  | 62.6  | 29.5  | 40.1  |      12.70      |  337.8   |
> |  √  |      √     |   √   |  √  |     |         | 0.978  | 77.3  | 76.3  | 76.8  | 0.796  | 70.3  | 63.6  | 66.8  |  0.983   | 93.6  | 87.3  | 90.3  | 0.671  | 63.7  | 30.6  | 41.3  |      12.81      |  347.6   |
> |  √  |      √     |   √   |  √  |  √  |         | 0.979  | 77.2  | 76.3  | 76.7  | 0.797  | 70.1  | 63.7  | 66.7  |  0.981   | 94.7  | 81.4  | 87.5  | 0.670  | 63.7  | 30.1  | 40.9  |      12.92      |  357.4   |
> |  √  |      √     |   √   |  √  |  √  |    √    | 0.979  | 77.2  | 76.1  | 76.6  | 0.794  | 69.9  | 63.6  | 66.6  |  0.980   | 93.4  | 87.1  | 90.1  | 0.670  | 63.6  | 30.3  | 41.0  |      13.08      |  367.2   |
>
> **Questions 1:Can you provide more detailed technical information about the consistency loss formulation in CPC? It would be helpful to see visual examples of how CPC affects the prompt generation process and its impact on the final detection results.**
>
>    Thank you for your constructive suggestions. We have supplemented explanations in the following two aspects:
>    - **Technical Details of CPC:** Refer to the response to Weakness 1.
>    - **Impact of CPC on Prompt Generation and Final Detection Results:** Refer to the response to Weakness 1.
>
> **Questions 2:Why were certain recent methods excluded from the comparison? Have you considered comparing with methods published in 2024 or other SAM-based approaches?**
>
>
>    Thank you for your constructive suggestions. We have provided explanations from the following two aspects:
>    - **Comparison with Important Recent Works from 2024:** Refer to the response to Weakness 2.
>    - **Comparison with Other Foundation Model-Based Methods:** Refer to the response to Weakness .
>
> **Questions 3:How does IMDPrompter handle images from different domains in practical applications? What mechanisms could be added to make the model more adaptive to distribution shifts?**
>
>    Thank you for your constructive suggestions. Refer to the response to Weakness 3.
>
> **Questions 4:Can you provide comprehensive ablation studies showing the contribution of each view and how different view combinations affect the model's performance? This should include an analysis of the computational overhead associated with each additional view.**
>
>    Thank you for your constructive suggestions. We have provided explanations from the following two aspects:
>    - **Ablation Studies on the Contribution of Each View:** Refer to the response to Weakness 4.
>    - **Computational Trade-Offs of Different View Combinations:** Refer to the response to Weakness 4.

---

> > ### Author Response · Authors · 2024-11-24
> >
> > **Questions 5:Could you explain the rationale behind choosing SRM, Bayer, and Noiseprint specifically as noise perspectives? Have experiments been conducted with other noise perspectives, and if so, what were the results?**
> >
> >    Thank you for your constructive suggestions. We have provided explanations from the following three aspects:
> >    - **Criteria for Selecting Types of Image Processing Information:** Refer to the response to Weakness 5.
> >    - **Reasons for Choosing SRM, Bayar, Noiseprint, and RGB as Prompt Views:** Refer to the response to Weakness 5.
> >    - **Experimental Results with Other Noise Views:** Refer to the response to Weakness 5.

---

> > > ### Comment · Reviewer_tXob · 2024-11-25
> > >
> > > Thanks for your hardworking response. I have updated my score

---

> > > > ### Author Response · Authors · 2024-12-01
> > > >
> > > > Dear Reviewer tXob,
> > > >
> > > > Thank you for your valuable feedback. We hope that our responses have sufficiently addressed your previous concerns. We noticed that you updated the score to 5 (marginally below the acceptance threshold). Should you have any additional comments or questions, we would be happy to hear them. If your concerns have been adequately resolved, we kindly hope that you might consider providing a score more favorable towards acceptance.
> > > >
> > > > Sincerely,
> > > > The Authors

---

> > > > > ### Comment · Reviewer_tXob · 2024-12-02
> > > > >
> > > > > Actually, I have the same doubt about the authenticity of experiment results as Reviewer z9xA. But considering your exhaustive response, I raise the score to 5 with negative trend.

---

> > > > > > ### Author Response · Authors · 2024-12-04
> > > > > >
> > > > > > ### Dear Reviewer tXob,
> > > > > >
> > > > > > Thank you for your valuable feedback. Regarding the experimental results on the CoCoGlide dataset, the following explanation is provided:
> > > > > >
> > > > > > Referring to the original data in **Trufor (CVPR 2023)** [1], the performance of **Trufor** and **CATNetv2** trained on CASIA v2, evaluated on the CoCoGlide dataset, is as follows:
> > > > > >
> > > > > > | Methods      | CoCoGlide | P-F1 (best) | P-F1 (fixed) | I-AUC | I-Acc |
> > > > > > |--------------|-----------|-------------|--------------|-------|-------|
> > > > > > | CAT-Net v2   | 0.603     | 0.434       | 0.667        | 0.580 |       |
> > > > > > | Trufor       | 0.720     | 0.523       | 0.752        | 0.639 |       |
> > > > > >
> > > > > > The **IMDPrompter**'s supplementary material (Table 18) uses the same experimental settings as Trufor and achieves similar performance.
> > > > > >
> > > > > > Referring to the original data in **UnionFormer (CVPR 2024)** [2], the performance of **Trufor**, **CATNetv2**, and **UnionFormer** trained on CASIA v2 and five additional datasets, evaluated on the CoCoGlide dataset, is as follows:
> > > > > >
> > > > > > | Methods      | CoCoGlide | P-F1 (best) | P-F1 (fixed) | I-AUC | I-Acc |
> > > > > > |--------------|-----------|-------------|--------------|-------|-------|
> > > > > > | CAT-Net v2   | 0.603     | 0.434       | 0.667        | 0.580 |       |
> > > > > > | Trufor       | 0.720     | 0.523       | 0.752        | 0.639 |       |
> > > > > > | UnionFormer  | 0.742     | 0.536       | 0.797        | 0.682 |       |
> > > > > >
> > > > > > The experiments conducted during the rebuttal process, using the same experimental settings as UnionFormer, show similar performance.
> > > > > >
> > > > > > In summary, the experimental results reported in both the original manuscript and the rebuttal regarding CoCoGlide are aligned with the results from **Trufor (CVPR 2023)** and **UnionFormer (CVPR 2024)** under the same experimental settings.
> > > > > >
> > > > > > We greatly appreciate your attention and recognition of our research. Your concerns about the experimental details are fully understood. Both the manuscript and rebuttal thoroughly describe the experimental design, analysis, results, as well as the data analysis and processing process. If there are any specific questions regarding the data, further clarification will be provided. Additionally, we commit to making all code publicly available after the paper is accepted.
> > > > > >
> > > > > > #### References
> > > > > > [1] Guillaro, F., Cozzolino, D., Sud, A., Dufour, N., & Verdoliva, L. (2023). Trufor: Leveraging all-round clues for trustworthy image forgery detection and localization. In *Proceedings of the IEEE/CVF conference on computer vision and pattern recognition* (pp. 20606-20615).
> > > > > >
> > > > > > [2] Li, S., Ma, W., Guo, J., Xu, S., Li, B., & Zhang, X. (2024). UnionFormer: Unified-Learning Transformer with Multi-View Representation for Image Manipulation Detection and Localization. In *Proceedings of the IEEE/CVF Conference on Computer Vision and Pattern Recognition* (pp. 12523-12533).

---

### Official Review · Reviewer_tfmM · 2024-11-02

**Soundness:** 3
**Presentation:** 3
**Contribution:** 3
**Rating:** 6
**Confidence:** 4

**Summary:**

This work introduces IMDPrompter, a novel cross-view prompt learning framework based on the Segment Anything Model (SAM) to automate detection and localization in image manipulation tasks, overcoming SAM’s reliance on manual prompts and enhancing cross-dataset generalization through cross-view perceptual learning techniques.

**Strengths:**

1. Innovative Application: The paper applies SAM  to the underexplored area of image manipulation detection, which extends SAM's use case beyond traditional segmentation tasks. The proposal of a cross-view automated prompt learning paradigm with IMDPrompter is unique and addresses the challenges specific to manipulation detection tasks.

2. Automation of Prompt Generation: One of the standout innovations is the elimination of SAM's reliance on manual prompts. The introduction of modules like Optimal Prompt Selection (OPS) and Cross-View Prompt Consistency (CPC) strengthens SAM’s utility by automating the prompt generation, potentially making SAM more accessible for manipulation detection.

3. Robustness and Generalizability: IMDPrompter's multi-view approach—integrating RGB, SRM, Bayer, and Noiseprint—demonstrates enhanced generalization, particularly on out-of-domain datasets. The ablation studies further substantiate the contributions of each module, which supports the validity of the multi-view and prompt-learning design.

4. Strong Experimental Validation: The model shows significant improvements in image-level and pixel-level metrics across multiple datasets (CASIA, Columbia, IMD2020, etc.), indicating its robustness. The experimental setup includes various metrics (AUC, F1-scores), highlighting the model’s strengths compared to prior approaches.

**Weaknesses:**

1. Complexity and Computational Cost: IMDPrompter’s architecture includes multiple modules (OPS, CPC, CFP, PMM), each introducing additional computational overhead. The increased complexity may impact the model’s efficiency, potentially limiting real-world deployment, especially for large-scale or time-sensitive tasks. Maybe we can provide computational complexity analysis or runtime comparisons with existing methods.

2. Limited Modality Discussion: While IMDPrompter is tested on a range of datasets for image manipulation, it would be beneficial if the authors discussed the potential application of this approach to other domains or modalities, such as video manipulation, to establish broader applicability. You can discuss particular challenges or modifications that would be needed to apply IMDPrompter to video manipulation detection.

3. The current framework provides limited insight into how each view or prompt contributes to the final decision-making process. Adding visualizations or more detailed explanations on how SAM’s interpretability might translate into manipulation detection could improve the work’s practical value.  For example, you can add the ablation studies showing the impact of each view, or visualizations of the learned prompts for different types of manipulations.

4. Reliance on Specific Views: The model’s reliance on SRM, Bayer, and Noiseprint views may limit its utility across other manipulation detection types that do not exhibit these specific signal properties. Further exploration into the model's adaptability to new types of data without these views might be necessary. I think you can discuss or demonstrate how IMDPrompter might be adapted to work with different types of views or features. Maybe you can also add an experiment with a subset of the current views to assess the model's flexibility.

**Questions:**

Real-World Deployment: Has IMDPrompter been tested in real-world settings, where variations in lighting, compression, and manipulation styles may further challenge the model’s robustness?

---

> ### Author Response · Authors · 2024-11-24
>
> **Weakness 1. Complexity and Computational Cost: IMDPrompter's architecture consists of multiple modules (OPS, CPC, CFP, PMM), each contributing to additional computational overhead. The increased complexity may impact the model's efficiency, potentially limiting real-world deployment, particularly for large-scale or time-sensitive tasks. To address this, it might be beneficial to provide a computational complexity analysis or runtime comparison with existing methods.**
>
> Thank you for your constructive suggestion. We analyzed the model's complexity from the following two perspectives:
>
> **Minimal Impact of Added Complexity Compared to the Baseline Method:** We developed a baseline method for image manipulation detection called SAM-IMD, which is based on SAM and uses only the RGB view to generate prompts online via an FCN network. We then progressively added multiple views and modules such as OPS, CPC, SAF, and PMM. Since all prompt views were implemented using an FCN network based on MobileNet, we observed a 0.165 increase in I-AUC and a 22.8% increase in P-F1, with only a 0.52-hour increase in training time and a 9.72% increase in parameter count.
>
>    | Baseline | Training Time (h) | Params (M) | Noiseprint | Bayar | SRM | OPS | CPC | SAF | PMM | I-AUC | P-F1 |
>    |----------|-------------------|------------|------------|-------|-----|-----|-----|-----|------|-------|------|
>    | SAM-IMD  | 12.29             | 316.8      |            |       |     |     |     |     |      | 0.631 | 39.8 |
>    |          | 12.43             | 326.6      | √          |       |     |     |     |     |      | 0.676 | 45.9 |
>    |          | 12.55             | 336.4      | √          | √     |     |     |     |     |      | 0.713 | 48.1 |
>    |          | 12.64             | 346.2      | √          | √     | √   |     |     |     |      | 0.731 | 51.7 |
>    |          | 12.71             | 346.2      | √          | √     | √   | √   |     |     |      | 0.752 | 52.6 |
>    |          | 12.72             | 346.2      | √          | √     | √   | √   | √   |     |      | 0.771 | 55.8 |
>    |          | 12.79             | 347.1      | √          | √     | √   | √   | √   | √   |      | 0.784 | 61.4 |
>    |          | 12.81             | 347.6      | √          | √     | √   | √   | √   | √   | √    | 0.796 | 63.6 |
>
> **Lightweight SAM Architecture for a More Compact IMDPrompter\*:** We implemented a lightweight version of IMDPrompter* based on MobileSAM. Under similar training time and parameter conditions as Trufor, IMDPrompter* achieved notable improvements, with an increase of 0.009 in I-AUC and 25.4% in P-F1.
>
>    |               | Training Time (h) | Params (M) | I-AUC  | P-F1  |
>    |---------------|-------------------|------------|--------|-------|
>    | ManTra-Net    | 13.72             | 1009.7     | 0.500  | 15.5  |
>    | MVSS-Net      | 5.16              | 160.0      | 0.731  | 45.2  |
>    | Trufor        | 4.71              | 90.1       | 0.770  | 19.9  |
>    | IMDPrompter*  | 4.76              | 85.8       | 0.779  | 45.3  |
>
> In summary, the significant improvement in detection accuracy achieved with a slight increase in complexity presents a favorable trade-off.

---

> ### Author Response · Authors · 2024-11-24
>
> **Weakness 2. Limited Modality Discussion: While IMDPrompter has been tested on a variety of datasets for image manipulation, it would be beneficial for the authors to discuss the potential application of this approach to other domains or modalities, such as video manipulation, to establish broader applicability. You could discuss specific challenges or necessary modifications to apply IMDPrompter to video manipulation detection.**
>
> Thank you for your constructive suggestion. We provide a detailed explanation from the following two perspectives:
>
> **Experiments with Subsets of the Current Views:** We found that when only a single view was used, the Noiseprint view achieved the best performance. Additionally, we observed that using the RGB view along with any semantic-agnostic view could achieve state-of-the-art (SOTA) performance across most metrics. Based on this, we concluded that for new data types, such as videos, employing a semantic-related view (e.g., RGB view) along with a semantic-agnostic view (e.g., Noiseprint view) could effectively facilitate manipulation detection.
>
>    | RGB | Noiseprint | Bayar | SRM | CASIA | COVER | Columbia | IMD | Training Time (h) | Params (M) | I-AUC | I-F1 | P-F1 | C-F1 |
>    |-----|------------|-------|-----|-------|-------|----------|-----|------------------|------------|-------|------|------|------|
>    | √   |            |       |     | 0.856 | 73.6  | 69.6     | 71.5| 12.31            | 318.10     | 0.556 | 19.6 | 38.6 | 26.0 |
>    |     | √          |       |     | 0.903 | 75.9  | 73.6     | 74.7| 12.31            | 318.10     | 0.761 | 62.9 | 58.7 | 60.7 |
>    |     |            | √     |     | 0.863 | 74.3  | 70.6     | 72.4| 12.31            | 318.10     | 0.706 | 61.3 | 56.3 | 58.7 |
>    |     |            |       | √   | 0.896 | 75.1  | 71.6     | 73.3| 12.31            | 318.10     | 0.713 | 59.6 | 55.6 | 57.5 |
>    | √   | √          |       |     | 0.941 | 76.9  | 75.2     | 76.0| 12.56            | 327.90     | 0.781 | 66.8 | 61.9 | 64.3 |
>    | √   |            | √     |     | 0.936 | 76.1  | 74.6     | 75.3| 12.56            | 327.90     | 0.769 | 65.7 | 60.8 | 63.2 |
>    | √   |            |       | √   | 0.937 | 76.4  | 74.5     | 75.4| 12.56            | 327.90     | 0.775 | 65.9 | 61.5 | 63.6 |
>    | √   | √          | √     |     | 0.962 | 77.1  | 75.9     | 76.5| 12.70            | 337.80     | 0.788 | 68.7 | 62.4 | 65.4 |
>    | √   | √          | √     | √   | 0.978 | 77.3  | 76.3     | 76.8| 12.81            | 347.60     | 0.796 | 70.3 | 63.6 | 66.8 |
>
> **Adapting IMDPrompter for Video Manipulation Detection:** To adapt IMDPrompter to video data, which we refer to as IMDPrompter v2, we propose the following modifications for improved adaptability:
>    a. Use SAM2[1] as the backbone model, capable of performing general video segmentation.
>    b. Use a semantic-related view (e.g., RGB view) and at least one noise-based view (e.g., Noiseprint) to generate prompts for SAM2[1].
>    c. Apply RNN, CNN, LSTM, or other similar modules to perform temporal modeling of features between adjacent frames from selected views.
>    d. Combine multi-view prompt information to guide SAM2[1] in video manipulation detection, while introducing CPC for cross-view consistency enhancement, OPS for optimal prompt selection, SAF for cross-view perceptual learning, and PMM to mix various types of prompts.
>
> [1] Ravi, Nikhila, et al. "Sam 2: Segment anything in images and videos." arXiv preprint arXiv:2408.00714 (2024).

---

> ### Author Response · Authors · 2024-11-24
>
> **Weakness 3. Limited Insight into View Contributions: The current framework provides limited insight into how each view or prompt contributes to the final decision-making process. Adding visualizations or more detailed explanations of how SAM’s interpretability might translate into manipulation detection could enhance the practical value of the work. For example, you could include ablation studies demonstrating the impact of each view or visualizations of the learned prompts for different types of manipulations.**
>
> Thank you for your constructive suggestion. Below is our detailed response:
>
> 1. **Ablation Studies on View Combinations:** As presented in Table 5 of the main text, we conducted ablation studies on different combinations of views. Initially, when each of the four views was used individually, we found that the Noiseprint view alone resulted in the highest detection accuracy. Furthermore, when we successively added the Noiseprint, SRM, and Bayar views on top of the RGB view, there were significant improvements in detection accuracy.
>
> 2. **Proportion of Times Each View Was Selected as the Optimal View:** According to Table 17 in the supplementary materials, we also analyzed the proportion of times each of the four views was selected as the optimal view. We observed that the Noiseprint view was chosen most frequently, highlighting its crucial role in the detection process. In contrast, the RGB view had the lowest selection rate, suggesting that for image manipulation detection tasks, higher detection accuracy relies heavily on the inclusion of semantic-agnostic views, such as SRM, Bayar, and Noiseprint.
>
> 3. **F1 Scores of Individual Prompt View Masks and the Final SAM Output Mask:** As shown in the table below, we compared the F1 scores of the masks generated from individual prompt views with the final mask output by SAM. The F1 score of the SAM output was significantly higher than those of the four prompt views, which demonstrates the substantial benefit of SAM's large-scale pre-training for image manipulation detection. Moreover, the F1 score for the Noiseprint view mask was higher than those of the other three views, indicating that the Noiseprint view provided the most optimal prompt.
>
>    | **RGB** | **SRM** | **Bayar** | **Noiseprint** | **SAM Output** |
>    |---------|---------|-----------|----------------|----------------|
>    | 65.7    | 66.1    | 65.4      | 67.9           | 76.8           |
>
> 4. **Visualization of Masks for Each Prompt View:**
>    In the revised version of our paper, we will include visualizations of the masks generated from each of the four prompt views for different types of manipulations.

---

> ### Author Response · Authors · 2024-11-24
>
> **Weakness 4. Reliance on Specific Views: The model’s reliance on SRM, Bayer, and Noiseprint views may limit its utility for detecting manipulations that do not exhibit these specific signal properties. Further exploration into the model's adaptability to new data types without these views might be necessary. You might consider discussing or demonstrating how IMDPrompter can be adapted to work with different types of views or features. Additionally, you could add an experiment using a subset of the current views to assess the model's flexibility.**
>
> Thank you for your constructive suggestion. We have provided a detailed analysis from the following two perspectives:
>
> **Experiments with Subsets of the Current Views:** We found that when using only a single view, the Noiseprint view performed the best. Furthermore, when we used both the RGB view and any semantic-agnostic view, we were able to achieve state-of-the-art (SOTA) performance for most metrics. Therefore, for novel data types (such as videos), we concluded that using one semantic-related view (e.g., RGB) along with one semantic-agnostic view (e.g., Noiseprint) would be effective for manipulation detection.
>
>    | RGB  | Noiseprint | Bayar | SRM | CASIA | COVER | Columbia | IMD | Training Time (h) | Params (M) |
>    |------|------------|-------|-----|-------|-------|----------|-----|-------------------|------------|
>    | √    |            |       |     | 0.856 | 73.6  | 69.6     | 71.5| 12.31             | 318.10     |
>    |      | √          |       |     | 0.903 | 75.9  | 73.6     | 74.7| 12.31             | 318.10     |
>    |      |            | √     |     | 0.863 | 74.3  | 70.6     | 72.4| 12.31             | 318.10     |
>    |      |            |       | √   | 0.896 | 75.1  | 71.6     | 73.3| 12.31             | 318.10     |
>    | √    | √          |       |     | 0.941 | 76.9  | 75.2     | 76.0| 12.56             | 327.90     |
>    | √    |            | √     |     | 0.936 | 76.1  | 74.6     | 75.3| 12.56             | 327.90     |
>    | √    |            |       | √   | 0.937 | 76.4  | 74.5     | 75.4| 12.56             | 327.90     |
>    | √    | √          | √     |     | 0.962 | 77.1  | 75.9     | 76.5| 12.70             | 337.80     |
>    | √    | √          | √     | √   | 0.978 | 77.3  | 76.3     | 76.8| 12.81             | 347.60     |
>
> **Adapting IMDPrompter for New Data Types:**
>    Specifically, when adapting IMDPrompter for video manipulation detection (resulting in IMDPrompter v2), we can make the following adjustments for better adaptability:
>    - a. Use **SAM2[1]** (capable of general video segmentation) as the backbone model.
>    - b. Generate prompts for SAM2[1] using an RGB view and at least one noise view (e.g., Noiseprint).
>    - c. Utilize **RRN**, **CNN**, **LSTM**, or other modules to capture and represent temporal features between adjacent frames across all views..
>    - d. Combine multi-view prompt information to detect video manipulation (further improve cross-view consistency using **CPC**, select the optimal prompt with **OPS**, enable cross-view perceptual learning with **SAF**, and mix different types of prompt information using **PMM**).
>
> **Question 1. Real-World Deployment: Has IMDPrompter been tested in real-world scenarios where variations in lighting, compression, and manipulation styles could further challenge the model's robustness?**
>
> **Robustness Analysis Against Gaussian Blur and JPEG Compression:** As shown in Figure 5 of the main text, we tested the performance of IMDPrompter, MVSS-Net, and other methods under different levels of JPEG compression and Gaussian blur. We found that IMDPrompter achieved the best performance in all settings, demonstrating good robustness against both Gaussian blur and JPEG compression.
>
> [1] Ravi, Nikhila, et al. "Sam 2: Segment anything in images and videos." arXiv preprint arXiv:2408.00714 (2024).

---

> ### Author Response · Authors · 2024-11-24
>
> **Robustness Analysis Against Lighting and WEBP Compression:** We also evaluated IMDPrompter and MVSS-Net under various lighting and WEBP compression conditions, as shown in the table below. IMDPrompter consistently achieved the best performance, demonstrating robustness across different lighting and WEBP compression scenarios.
>
> |      P-F1     |       |       | Lighting |       |       |       |       |       | WEBP compress  |       |       |       |
> |:-------------:|:-----:|:-----:|:--------:|:-----:|:-----:|:-----:|:-----:|:-----:|:--------------:|:-----:|:-----:|:-----:|
> |     Method    |  -75  |  -50  |    -25   |   0   |   25  |   50  |   100   |   90   |       80       |   70  |   60  |   50  |
> |   H-LSTM[57]  | 14.3  | 14.7  |   15.4   | 15.4  | 15.3  | 15.0  | 15.4  | 13.2  |      12.1      | 10.8  | 10.6  | 10.1  |
> | ManTra-Net[6] | 15.1  | 15.2  |   15.3   | 15.5  | 15.4  | 15.2  | 15.5  | 11.8  |      11.2      | 11.8  | 11.5  | 11.1  |
> |   CR-CNN[17]  | 39.0  | 39.4  |   39.7   | 40.4  | 40.1  | 39.0  | 40.4  | 22.9  |      21.9      | 20.9  | 20.3  | 19.8  |
> |  GSR-Net[54]  | 35.3  | 36.1  |   36.5   | 37.4  | 37.3  | 37.0  | 37.4  | 30.4  |      26.3      | 23.3  | 20.8  | 18.9  |
> |    SPAN[16]   | 17.4  | 17.4  |   17.7   | 18.2  | 18.1  | 17.7  | 18.2  | 10.5  |      10.4      | 10.0  |  9.4  |  9.5  |
> |  CAT-Net[58]  | 12.5  | 12.8  |   13.2   | 13.5  | 13.2  | 12.8  | 13.5  |  9.3  |      9.2       |  6.9  |  6.0  |  5.9  |
> |  MVSS-Net[59] | 43.3  | 44.3  |   45.0   | 45.1  | 44.6  | 44.2  | 45.1  | 37.0  |      35.9      | 32.9  | 21.1  | 17.2  |
> |  IMDPrompter  | 74.3  | 74.8  |   76.0   | 76.3  | 75.5  | 74.8  | 76.3  | 71.2  |      70.2      | 67.5  | 55.7  | 50.9  |
>
> |      I-F1     |       |       | Lighting |       |       |       |       |       | WEBP compress |       |       |       |
> |:-------------:|:-----:|:-----:|:--------:|:-----:|:-----:|:-----:|:-----:|:-----:|:-------------:|:-----:|:-----:|:-----:|
> |     Method    |  -75  |  -50  |    -25   |   0   |   25  |   50  |   100   |   90   |       80       |   70  |   60  |   50  |
> |   H-LSTM[57]  |  0.0  |  0.0  |   0.0    |  0.0  |  0.0  |  0.0  |  0.0  |  0.0  |      0.0      |  0.0  |  0.0  |  0.0  |
> | ManTra-Net[6] |  0.0  |  0.0  |   0.0    |  0.0  |  0.0  |  0.0  |  0.0  |  0.0  |      0.0      |  0.0  |  0.0  |  0.0  |
> |   CR-CNN[17]  | 23.3  | 23.7  |   24.3   | 24.3  | 24.1  | 23.5  | 24.3  | 14.5  |     14.1      | 13.9  | 12.4  | 11.4  |
> |  GSR-Net[54]  |  0.0  |  0.0  |   0.0    |  0.0  |  0.0  |  0.0  |  0.0  |  0.0  |      0.0      |  0.0  |  0.0  |  0.0  |
> |    SPAN[16]   |  0.0  |  0.0  |   0.0    |  0.0  |  0.0  |  0.0  |  0.0  |  0.0  |      0.0      |  0.0  |  0.0  |  0.0  |
> |  CAT-Net[58]  | 37.0  | 37.5  |   37.8   | 38.0  | 37.2  | 36.4  | 38.0  | 12.1  |     11.0      | 10.6  | 10.4  | 10.7  |
> |  MVSS-Net[59] | 72.4  | 74.3  |   74.8   | 75.9  | 73.7  | 72.7  | 75.9  | 74.1  |     73.1      | 70.9  | 66.6  | 64.0  |
> |  IMDPrompter  | 74.7  | 74.9  |   76.8   | 77.3  | 75.9  | 74.0  | 77.3  | 76.8  |     75.3      | 74.5  | 70.1  | 67.5  |
>
> |     Com-F1    |       |       | Lighting |       |       |       |       |       | WEBP compress |       |       |       |
> |:-------------:|:-----:|:-----:|:--------:|:-----:|:-----:|:-----:|:-----:|:-----:|:-------------:|:-----:|:-----:|:-----:|
> |     Method    |  -75  |  -50  |    -25   |   0   |   25  |   50  |  100   |   90   |       80       |   70  |   60  |   50  |
> |   H-LSTM[57]  |  0.0  |  0.0  |   0.0    |  0.0  |  0.0  |  0.0  |  0.0  |  0.0  |      0.0      |  0.0  |  0.0  |  0.0  |
> | ManTra-Net[6] |  0.0  |  0.0  |   0.0    |  0.0  |  0.0  |  0.0  |  0.0  |  0.0  |      0.0      |  0.0  |  0.0  |  0.0  |
> |   CR-CNN[17]  | 29.3  | 29.6  |   30.1   | 30.3  | 30.1  | 29.4  | 30.3  | 17.8  |     17.2      | 16.7  | 15.4  | 14.5  |
> |  GSR-Net[54]  |  0.0  |  0.0  |   0.0    |  0.0  |  0.0  |  0.0  |  0.0  |  0.0  |      0.0      |  0.0  |  0.0  |  0.0  |
> |    SPAN[16]   |  0.0  |  0.0  |   0.0    |  0.0  |  0.0  |  0.0  |  0.0  |  0.0  |      0.0      |  0.0  |  0.0  |  0.0  |
> |  CAT-Net[58]  | 18.9  | 19.1  |   19.6   | 19.9  | 19.5  | 19.0  | 19.9  | 10.5  |     10.0      |  8.4  |  7.6  |  7.6  |
> |  MVSS-Net[59] | 54.6  | 55.5  |   56.2   | 56.6  | 55.6  | 55.0  | 56.6  | 49.3  |     48.2      | 44.9  | 32.0  | 27.1  |
> |  IMDPrompter  | 73.8  | 74.8  |   76.4   | 76.8  | 75.7  | 74.4  | 76.8  | 73.9  |     72.7      | 70.8  | 62.1  | 58.1  |

---

> ### Author Response · Authors · 2024-11-24
>
> - **Robustness Analysis Against a Wider Range of Tampering Styles:** As detailed in the table below, we utilized the same combined training set as UnionFormer [1] and evaluated performance on each test set. These test sets included various types of image manipulation, such as copy-move (copying and moving elements from one region to another within the same image), splicing (copying elements from one image and pasting them onto another), inpainting (removing unwanted elements), and image manipulations using diffusion models. Our method demonstrated state-of-the-art (SOTA) performance, validating its effectiveness against additional tampering techniques.
>
> ### Pixel-level F1 Experimental Analysis
>
> | **Model**        | **Optimal Threshold** |                       |                       |                       |                       | **AVG** | **Fixed Threshold (0.5)** |                       |                       |                       |                       | **AVG** |
> |------------------|-----------------------|-----------------------|-----------------------|-----------------------|-----------------------|---------|---------------------------|-----------------------|-----------------------|-----------------------|-----------------------|---------|
> |                  | Columbia              | Coverage              | CASIA                 | NIST                  | CoCoGlide             |         | Columbia                  | Coverage              | CASIA                 | NIST                  | CoCoGlide             |         |
> | **CAT-Net v2**   | 92.3                  | 58.2                  | 85.2                  | 41.7                  | 60.3                  | 67.5    | 85.9                      | 38.1                  | 75.2                  | 30.8                  | 43.4                  | 54.7    |
> | **Trufor**       | 91.4                  | 73.5                  | 82.2                  | 47.0                  | 72.0                  | 73.2    | 85.9                      | 60.0                  | 73.7                  | 39.9                  | 52.3                  | 62.4    |
> | **UnionFormer[1]**  | 92.5                  | 72.0                  | 86.3                  | 48.9                  | 74.2                  | 74.8    | 86.1                      | 59.2                  | 76.0                  | 41.3                  | 53.6                  | 63.2    |
> | **IMDPrompter**  | 92.7                  | 74.0                  | 87.1                  | 50.1                  | 74.0                  | 75.6    | 86.7                      | 60.3                  | 76.8                  | 42.4                  | 52.9                  | 63.8    |
>
> ### Image-level Detection Experimental Analysis
>
> | **Model**        | **Image-level AUC** |                       |                       |                       |                       | **AVG** | **Accuracy**               |                       |                       |                       |                       | **AVG** |
> |------------------|---------------------|-----------------------|-----------------------|-----------------------|-----------------------|---------|----------------------------|-----------------------|-----------------------|-----------------------|-----------------------|---------|
> |                  | Columbia            | Coverage              | CASIA                 | NIST                  | CoCoGlide             |         | Columbia                   | Coverage              | CASIA                 | NIST                  | CoCoGlide             |         |
> | **CAT-Net v2**   | 0.977               | 0.680                 | 0.942                 | 0.750                 | 0.667                 | 0.803   | 0.803                      | 0.635                 | 0.838                 | 0.597                 | 0.580                 | 0.691   |
> | **Trufor**       | 0.996               | 0.770                 | 0.916                 | 0.760                 | 0.752                 | 0.839   | 0.984                      | 0.680                 | 0.813                 | 0.662                 | 0.639                 | 0.756   |
> | **UnionFormer[1]**  | 0.998               | 0.783                 | 0.951                 | 0.793                 | 0.797                 | 0.864   | 0.979                      | 0.694                 | 0.843                 | 0.680                 | 0.682                 | 0.776   |
> | **IMDPrompter**  | 0.998               | 0.791                 | 0.962                 | 0.801                 | 0.801                 | 0.871   | 0.984                      | 0.703                 | 0.851                 | 0.682                 | 0.680                 | 0.780   |

---

> > ### Author Response · Authors · 2024-11-24
> >
> > ### Pixel-level AUC Experimental Analysis
> >
> > | **Model**        | Columbia | Coverage | CASIA | NIST | IMD  | **AVG** |
> > |------------------|----------|----------|-------|------|------|---------|
> > | **Trufor**       | 0.947    | 0.925    | 0.957 | 0.877| -    | 0.927   |
> > | **UnionFormer[1]**  | 0.989    | 0.945    | 0.972 | 0.881| 0.860| 0.929   |
> > | **IMDPrompter**  | 0.990    | 0.948    | 0.978 | 0.890| 0.864| 0.934   |
> >
> > [1] **UnionFormer:** Unified-Learning Transformer with Multi-View Representation for Image Manipulation Detection and Localization

---

> ### Comment · Reviewer_tfmM · 2024-11-27
>
> Thank you for your comments. I modified my original rating.

---

### Official Review · Reviewer_t8Py · 2024-11-03

**Soundness:** 2
**Presentation:** 2
**Contribution:** 2
**Rating:** 6
**Confidence:** 3

**Summary:**

The paper addresses the application of the SAM to the domain of image manipulation detection. The paper proposes IMDPrompter, a cross-view automated prompt learning paradigm that extends SAM's capabilities for IMD tasks. The proposed method is evaluated on five datasets (CASIA, Columbia, COVER, IMD2020, and NIST16), demonstrating significant improvements over existing state-of-the-art methods in both in-distribution and out-of-distribution settings.

**Strengths:**

The paper is well-motivated. The authors use SAM to address the challenges in the image manipulation detection domain. The introduction of automated prompt learning and integration of semantic-agnostic features is innovative and extends the applicability of SAM to a new domain.

The design is straightforward and reasonable. Apart from RGB images, using multiple views provides more information for IMD.

Five datasets and ablation studies demonstrate the effectiveness of the proposed IMDPrompter.

**Weaknesses:**

1. The proposed method introduces several additional modules, encoders, and views, which increase the complexity of the model.

2. Although the combination of four additional modules increases the performance, the authors do not provide the mechanism or reason behind why they chose views of SRM, Bayar, and Noiseprint, and which type of image processing information would help the IMDPrompter generate a more precise mask.

3. There is no CFP in Figure 2, but it is described in the caption.

4. There are no details of the architecture of the SRM/RGB/Bayar/Noiseprint encoder and its computational cost.

5. While the paper acknowledges that relying solely on RGB information is insufficient for cross-dataset generalization, there is limited discussion on scenarios where the proposed semantic-agnostic views might also fail, such as advanced manipulation techniques that bypass noise-based detectors.

6. The number of abbreviations is too many, which may interrupt the experience and hinder understanding for readers not familiar with the notation.

7. It would be better to bold the best results in Table 3,4,5,6.

**Questions:**

Please see the weakness part 2, 3, 4, 5, 6, 7.

---

> ### Author Response · Authors · 2024-11-23
>
> **Weakness 1：The proposed method introduces several additional modules, encoders, and views, which increase the complexity of the model.**
>    - Thank you for your constructive suggestion. We analyzed the model's complexity from the following two perspectives:
>
>    - **Minimal impact of additional complexity introduced by multi-view and modules like OPS compared to the baseline method:**  We constructed a SAM-based baseline method for image manipulation detection named SAM-IMD (prompt views are generated online using an RGB-only FCN). Subsequently, we progressively introduced multiple views and modules such as OPS, CPC, SAF, and PMM for the baseline. Since the prompt views are implemented using MobileNet-based FCN, the additional training time was only 0.52 hours, and parameter count increased by 9.72%. This resulted in a significant improvement of 0.165 and 22.8% in I-AUC and P-F1, respectively.
>
>      | **Baseline** | **Training Time (h)** | **Params (M)** | **Noiseprint** | **Bayar** | **SRM** | **OPS** | **CPC** | **SAF** | **PMM** | **I-AUC** | **P-F1** |
>      |--------------|-----------------------|----------------|----------------|-----------|---------|---------|---------|---------|---------|-----------|---------|
>      | SAM-IMD      | 12.29                | 316.8          |                |           |         |         |         |         |         | 0.631     | 39.8    |
>      |              | 12.43                | 326.6          | √              |           |         |         |         |         |         | 0.676     | 45.9    |
>      |              | 12.55                | 336.4          | √              | √         |         |         |         |         |         | 0.713     | 48.1    |
>      |              | 12.64                | 346.2          | √              | √         | √       |         |         |         |         | 0.731     | 51.7    |
>      |              | 12.71                | 346.2          | √              | √         | √       | √       |         |         |         | 0.752     | 52.6    |
>      |              | 12.72                | 346.2          | √              | √         | √       | √       | √       |         |         | 0.771     | 55.8    |
>      |              | 12.79                | 347.1          | √              | √         | √       | √       | √       | √       |         | 0.784     | 61.4    |
>      |              | 12.81                | 347.6          | √              | √         | √       | √       | √       | √       | √       | 0.796     | 63.6    |
>
>    - **Lightweight SAM architecture enables a more efficient IMDPrompter\*:** Using MobileSAM, we developed a lightweight IMDPrompter\*. Under similar training time and parameter count conditions as Trufor, IMDPrompter* achieved a significant improvement of 0.009 and 25.4% in I-AUC and P-F1, respectively.
>
>      | **Model**     | **Training Time (h)** | **Params (M)** | **I-AUC** | **P-F1** |
>      |---------------|-----------------------|----------------|-----------|----------|
>      | ManTra-Net    | 13.72                | 1009.7         | 0.500     | 15.5     |
>      | MVSS-Net      | 5.16                 | 160.0          | 0.731     | 45.2     |
>      | Trufor        | 4.71                 | 90.1           | 0.770     | 19.9     |
>      | IMDPrompter*  | 4.76                 | 85.8           | 0.779     | 45.3     |
>
>    In conclusion, the substantial improvement in detection accuracy achieved with minimal additional complexity represents a favorable trade-off.

---

> > ### Author Response · Authors · 2024-11-23
> >
> > **Weakness 2：Although the combination of four additional modules improves performance, the authors do not provide the rationale for choosing views of SRM, Bayar, and Noiseprint, or explain which types of image processing information enhance IMDPrompter's mask generation precision.**
> >    - **Criteria for choosing image processing information types:**
> >      The SRM, Bayar, and Noiseprint views were selected to introduce semantic-agnostic features. Semantic-agnostic features are defined as:
> >      “Low-level artifacts caused by the in-camera acquisition process, such as the sensor, lens, color filter array, or JPEG quantization tables. For Image Manipulation Detection tasks, semantic-agnostic features highlight low-level artifact information, which differs significantly between manipulated and non-manipulated regions of an image. Common extraction methods include SRM, Bayar, DCT, and Noiseprint.”
> >      Therefore, views that emphasize low-level artifacts and show strong differences between manipulated and non-manipulated regions are ideal for this task.

---

> > > ### Author Response · Authors · 2024-11-23
> > >
> > > - **Reasons for selecting SRM, Bayar, Noiseprint, and RGB as prompt views:**
> > >      The selection of semantic-agnostic features must meet the following criteria:
> > >      a. Each position in the noise view must correspond one-to-one with the original image.
> > >      b. The view must emphasize low-level artifact information to maximize differences between manipulated and non-manipulated regions.
> > >      Considering these criteria and drawing on the experience of previous work (refer to Supplementary Material Table 8), we selected Noiseprint, SRM, Bayar, DCT, and Wavelet views. Through ablation experiments, we observed the following:
> > >      - Using only the Wavelet view performed the worst on in-domain datasets and only slightly better than RGB in cross-domain tests, indicating limited utility.
> > >      - The sequential addition of Noiseprint, Bayar, and SRM views resulted in significant improvements in detection quality and enhanced cross-dataset generalization capabilities.
> > >      - Adding DCT and Wavelet views did not further enhance performance.
> > >      Considering computational cost and detection accuracy, we selected RGB, Noiseprint, SRM, and Bayar as the final prompt views.'
> > >
> > > | RGB | Noiseprint | Bayar | SRM | DCT | Wavelet |  CASIA |       |       |       |  COVER |       |       |       | Columbia |       |       |       |   IMD  |       |       |       | training time/h | Params/M |
> > > |:---:|:----------:|:-----:|:---:|:---:|:-------:|:------:|:-----:|:-----:|:-----:|:------:|:-----:|:-----:|:-----:|:--------:|:-----:|:-----:|:-----:|:------:|:-----:|:-----:|:-----:|:---------------:|:--------:|
> > > |     |            |       |     |     |         |  I-AUC |  I-F1 |  P-F1 |  C-F1 |  I-AUC |  I-F1 |  P-F1 |  C-F1 |   I-AUC  |  I-F1 |  P-F1 |  C-F1 |  I-AUC |  I-F1 |  P-F1 |  C-F1 |                 |          |
> > > |  √  |            |       |     |     |         | 0.856  | 73.6  | 69.6  | 71.5  | 0.556  | 19.6  | 38.6  | 26.0  |  0.786   | 53.6  | 24.9  | 34.0  | 0.531  | 28.1  | 21.2  | 24.2  |      12.31      |  318.1   |
> > > |     |      √     |       |     |     |         | 0.903  | 75.9  | 73.6  | 74.7  | 0.761  | 62.9  | 58.7  | 60.7  |  0.946   | 88.6  | 83.5  | 86.0  | 0.636  | 59.9  | 27.9  | 38.1  |      12.31      |  318.1   |
> > > |     |            |   √   |     |     |         | 0.863  | 74.3  | 70.6  | 72.4  | 0.706  | 61.3  | 56.3  | 58.7  |  0.889   | 83.4  | 81.6  | 82.5  | 0.601  | 54.6  | 25.7  | 34.9  |      12.31      |  318.1   |
> > > |     |            |       |  √  |     |         | 0.896  | 75.1  | 71.6  | 73.3  | 0.713  | 59.6  | 55.6  | 57.5  |  0.906   | 84.6  | 79.3  | 81.9  | 0.593  | 56.3  | 24.3  | 33.9  |      12.31      |  318.1   |
> > > |     |            |       |     |  √  |         | 0.861  | 74.1  | 70.7  | 72.4  | 0.613  | 32.4  | 47.3  | 38.5  |  0.843   | 75.6  | 70.7  | 73.1  | 0.553  | 36.9  | 22.1  | 27.6  |      12.31      |  318.1   |
> > > |     |            |       |     |     |    √    | 0.746  | 66.7  | 62.1  | 64.3  | 0.576  | 23.6  | 41.3  | 30.0  |  0.791   | 54.9  | 26.4  | 35.7  | 0.560  | 30.9  | 23.6  | 26.8  |      12.31      |  318.1   |
> > > |  √  |      √     |       |     |     |         | 0.941  | 76.9  | 75.2  | 76.0  | 0.781  | 66.8  | 61.9  | 64.3  |  0.969   | 91.7  | 85.4  | 88.4  | 0.655  | 61.5  | 28.8  | 39.2  |      12.56      |  327.9   |
> > > |  √  |      √     |   √   |     |     |         | 0.962  | 77.1  | 75.9  | 76.5  | 0.788  | 68.7  | 62.4  | 65.4  |  0.977   | 92.4  | 86.5  | 89.4  | 0.660  | 62.6  | 29.5  | 40.1  |      12.70      |  337.8   |
> > > |  √  |      √     |   √   |  √  |     |         | 0.978  | 77.3  | 76.3  | 76.8  | 0.796  | 70.3  | 63.6  | 66.8  |  0.983   | 93.6  | 87.3  | 90.3  | 0.671  | 63.7  | 30.6  | 41.3  |      12.81      |  347.6   |
> > > |  √  |      √     |   √   |  √  |  √  |         | 0.979  | 77.2  | 76.3  | 76.7  | 0.797  | 70.1  | 63.7  | 66.7  |  0.981   | 94.7  | 81.4  | 87.5  | 0.670  | 63.7  | 30.1  | 40.9  |      12.92      |  357.4   |
> > > |  √  |      √     |   √   |  √  |  √  |    √    | 0.979  | 77.2  | 76.1  | 76.6  | 0.794  | 69.9  | 63.6  | 66.6  |  0.980   | 93.4  | 87.1  | 90.1  | 0.670  | 63.6  | 30.3  | 41.0  |      13.08      |  367.2   |
> > >
> > > **Weakness 3：There is no CFP in Figure 2, but it is described in the caption.**
> > >    Thank you for your suggestion. We will revise it to SAF in the updated version.
> > >
> > > **Weakness 4：There are no details of the architecture of the SRM/RGB/Bayar/Noiseprint encoder and its computational cost.**
> > >    The SRM, RGB, Bayar, and Noiseprint views all use a MobileNetV2-based FCN model (9.8M parameters, 38.4 GFLOPs) as the encoder. The FCN models for each view have consistent architecture but do not share parameters.

---

> ### Author Response · Authors · 2024-11-23
>
> **Weakness 5：While the paper acknowledges that relying solely on RGB information is insufficient for cross-dataset generalization, there is limited discussion on scenarios where the proposed semantic-agnostic views might also fail, such as advanced manipulation techniques that bypass noise-based detectors.**
>    Thank you for your suggestion. Below are some scenarios where IMDPrompter might fail:
>    - **Manipulated medical images:** IMDPrompter training datasets, such as CASIAv2 and CoCoGlide, are natural scene images. For medical images, which differ significantly in distribution, performance may degrade. Fine-tuning on medical datasets may be necessary.
>    - **Video manipulation:** Advanced video generation models, such as SORA, pose challenges as manipulation extends to video content. While this presents difficulties, SAM2[1] (Segment anything in images and videos) has extended capabilities to videos. Developing IMDPrompter v2, which is better suited for video manipulation detection, is an urgent research area.
>
> **Weakness 6：The number of abbreviations is too many, which may interrupt the reader’s understanding.**
>    Thank you for your suggestion. We will reduce unnecessary abbreviations in the revised version.
>
> **Weakness 7：It would be better to bold the best results in Tables 3, 4, 5, and 6.**
>    Thank you for your suggestion. We will bold the best results in Tables 3, 4, 5, and 6 in the revised version.
>
> [1] Ravi, Nikhila, et al. "Sam 2: Segment anything in images and videos." arXiv preprint arXiv:2408.00714 (2024).

---

> > ### Comment · Reviewer_t8Py · 2024-11-25
> >
> > Thank you for the comments. It helps clarify some problems. I maintain my original rating.

---

### Official Review · Reviewer_z9xA · 2024-11-03

**Soundness:** 3
**Presentation:** 3
**Contribution:** 3
**Rating:** 6
**Confidence:** 3

**Summary:**

The authors propose an automated prompt learning SAM-based method for image manipulation detection. Multiple views, such as RGB, SRM, Bayar, and Noiseprint, are utilized and integrated to generate auxiliary masks and bounding boxes for SAM. Meanwhile, many modules, such as Cross-view Feature Perception and Prompt Mixing modules, are proposed for mixing features for the Mask Decoder. Extensive results demonstrate the effectiveness of the proposed method.

**Strengths:**

1. The authors utilize SAM's zero-shot capabilities for image manipulation detection.
2. Multiple views, such as RGB, SRM, Bayar, and Noiseprint, are utilized and integrated to generate auxiliary masks and bounding boxes for SAM
3. Many modules, such as Cross-view Feature Perception and Prompt Mixing modules, are proposed for mixing features for the Mask Decoder.
4. Extensive results demonstrate the effectiveness of the proposed method.

**Weaknesses:**

1. Can the authors clarify why SAM is employed in image manipulation detection? The authors utilize four different views to generate masks and then bounding boxes, which serve as prompts for the Mask Decoder. To the best of my knowledge, the output masks of the Mask Decoder are significantly dependent on prompt accuracy. It is essential that the generated masks are sufficiently accurate. Can the authors provide the accuracy metrics for both the generated masks and the output masks from the Mask Decoder? Additionally, is it necessary to employ SAM?
2. Are the Mask Decoder and Prompt Encoder kept frozen during the training process?
3. In the PMM module, the feature F_{SAF}, which is encoded from images, is resized to the same shape as the output of the Prompt Encoder, F_{opt}, which is encoded based on coordinates. Could the authors elaborate on the motivation for this approach? The fusions of image embeddings and coordinate embeddings appears inconsistent.
4. The authors claim that the proposed OPS and CPC enhance alignment across views. Ideally, they utilize the CPC loss function to achieve prompt consistency. However, However, this claim lacks convincing evidence. Can the authors provide details on how the two proposed modules contribute to improved alignment?
5. The proposed method is trained only on the CASIAv2 dataset, while several other studies, such as CAT-Net v2 [1], TruFor [2], and UnionFormer [3], utilize additional datasets for training. In Table 1 and Table 2, the metrics are based on the CASIAv2 dataset without additional datasets. Can you explain why the method is only trained on the CASIAv2 dataset?
6. Table 2 presents numerous NaN values for Sensitivity, Specificity, and F1 scores related to TruFor. Can the authors provide the corresponding metrics?
7. Can you provide more recent methods for comparison? such as UnionFormer [3].

[1] Kwon, Myung-Joon, et al. "Learning jpeg compression artifacts for image manipulation detection and localization." International Journal of Computer Vision 130.8 (2022): 1875-1895.

[2] Guillaro, Fabrizio, et al. "Trufor: Leveraging all-round clues for trustworthy image forgery detection and localization." Proceedings of the IEEE/CVF conference on computer vision and pattern recognition. 2023.

[3] Li, Shuaibo, et al. "UnionFormer: Unified-Learning Transformer with Multi-View Representation for Image Manipulation Detection and Localization." Proceedings of the IEEE/CVF Conference on Computer Vision and Pattern Recognition. 2024.

**Questions:**

Please see the weakness.

---

> ### Author Response · Authors · 2024-11-23
>
> **Weakness 1：Can the authors clarify why SAM is employed in image manipulation detection?** The authors utilize four different views to generate masks and subsequently bounding boxes, which serve as prompts for the Mask Decoder. To the best of my knowledge, the output masks of the Mask Decoder heavily depend on the accuracy of these prompts. It is essential that the generated masks maintain sufficient accuracy. Can the authors provide accuracy metrics for both the generated masks and the final output masks from the Mask Decoder? Additionally, is SAM necessary for this process?
> - Thank you for your constructive suggestions. Our detailed explanation is as follows
> - **Ablation Study of SAM:** Referring to Table 6 in the main text, we conducted an ablation study on SAM. We constructed FCN+ (which does not use SAM but generates masks from multiple views). Our findings indicate that by combining information from multiple views, FCN+ achieves a significant performance improvement compared to FCN. However, there is still a substantial performance gap when compared to IMDPrompter. The reason for this is that SAM is pre-trained on the large-scale segmentation dataset SA-1B, which enables it to acquire generalizable prior knowledge. This prior knowledge helps IMDPrompter achieve more accurate image manipulation detection performance.
>
> - **Quality Analysis of Prompt View Masks and Final Output Masks:** Furthermore, we have included the F1 scores for the masks generated from the four prompt views as well as the final output masks. The details are provided in the following table.
> -
> | **RGB** | **SRM** | **Bayar** | **Noiseprint** | **SAM Output** |
> |---------|---------|-----------|----------------|----------------|
> | 65.7    | 66.1    | 65.4      | 67.9           | 76.8           |
>
> **Weakness 2：Are the Mask Decoder and Prompt Encoder kept frozen during the training process?**
>
> - The parameters of the Mask Decoder and Prompt Encoder are trainable. (In Figure 2 of the main text, the Image Encoder is marked as having frozen parameters, while other modules are not marked as frozen. To avoid confusion, we will explicitly indicate the trainable modules in the revised version.)
>
>
>
> **Weakness 3：In the PMM module, the feature \(F_{SAF}\), encoded from images, is resized to match the shape of \(F_{opt}\), the output from the Prompt Encoder based on coordinates. Could the authors elaborate on the motivation for this approach? The fusion of image embeddings and coordinate embeddings appears inconsistent.**
>
> - **Motivation:** Referring to the original SAM paper, SAM supports two types of prompts: Dense Prompt and Sparse Prompt. For Dense Prompt, the original SAM paper models features \(F_{Dense}\) by applying convolution operations on masks. We observed that \(F_{Dense}\) exhibits higher activation values for target regions and lower activation values for non-target regions. Similarly, \(F_{SAF}\) possesses these characteristics and provides more specific information for image manipulation detection during decoding. Therefore, we proposed PMM to fuse \(F_{SAF}\) and \(F_{opt}\).
>
> - **Fusion of image and coordinate embeddings:** As described in the Motivation section, \(F_{SAF}\) has higher activation values for target regions, offering a coarse localization of these areas. It also contains rich, specific information related to image manipulation detection, making it suitable for fusion with \(F_{opt}\) to generate more accurate prompts.

---

> > ### Author Response · Authors · 2024-11-23
> >
> > **Weakness 4：The authors claim that the proposed OPS and CPC modules enhance alignment across views. While CPC is intended to ensure prompt consistency, this claim lacks convincing evidence. Can the authors provide details on how these modules contribute to improved alignment?**
> >
> > - Thank you for your constructive suggestion. We have supplemented the paper with a more detailed ablation study on OPS and CPC:
> >
> > - To analyze OPS and CPC, we replaced OPS with a weighted average of the outputs from the four views, which served as an ensemble prompt for the decoding process.
> >
> >   - **Cosine similarity relative to the optimal prompt:** After applying OPS, the cosine similarity between individual view prompts and the optimal prompt did not increase significantly, indicating that OPS alone is insufficient for aligning with the optimal prompt. However, when OPS and CPC are used together, the cosine similarity between all views and the optimal view significantly increases (exceeding 0.95 in all cases). This demonstrates that combining OPS and CPC effectively aligns prompts from different views with the optimal prompt.
> >
> > |                | **RGB** | **SRM** | **Bayar** | **Noiseprint** |
> > |----------------|---------|---------|-----------|----------------|
> > | **Weighted Average** | 0.82    | 0.85    | 0.83      | 0.92           |
> > | **OPS**         | 0.81    | 0.86    | 0.83      | 0.99           |
> > | **OPS+CPC**     | 0.95    | 0.97    | 0.96      | 0.99           |
> >
> >
> >    - **F1 Scores Relative to Ground Truth (GT):** Furthermore, we evaluated the F1 scores of the prompt masks generated from each view against the ground truth (GT). Compared to the weighted average strategy, OPS effectively prevents the degradation of the optimal prompt by inaccurate prompts, thereby improving the overall image manipulation detection performance. However, we observed that using OPS alone does not lead to an improvement in the F1 scores of the prompt masks from individual views. This suggests that OPS alone is insufficient to enhance cross-view consistency. When OPS is combined with CPC, not only does the overall image manipulation detection performance improve, but the accuracy of the prompt masks from individual views also increases. This demonstrates that the combined use of OPS and CPC effectively enhances cross-view consistency, bringing each view closer to the optimal prompt.
> >
> > |                | **RGB** | **SRM** | **Bayar** | **Noiseprint** | **SAM Output** |
> > |----------------|---------|---------|-----------|----------------|----------------|
> > | **Weighted Average** | 42.9    | 58.1    | 54.3      | 61.7           | 73.8           |
> > | **OPS**         | 42.5    | 58.3    | 55.0      | 62.4           | 75.4           |
> > | **OPS+CPC**     | 65.7    | 66.1    | 65.4      | 67.9           | 76.8           |

---

> > > ### Author Response · Authors · 2024-11-23
> > >
> > > **Weakness 5：The proposed method is trained only on the CASIAv2 dataset, while several other studies, such as CAT-Net v2 [1], TruFor [2], and UnionFormer [3], utilize additional datasets for training. In Tables 1 and 2, the metrics are based solely on the CASIAv2 dataset without using additional datasets. Can the authors explain why the proposed method is trained only on the CASIAv2 dataset?**
> > >
> > > - Thank you for the constructive suggestion. Our detailed explanation is as follows:
> > >
> > >   - **Alignment with MVSS-Net experimental settings:** MVSS-Net uses CASIAv2 as the training dataset. we followed the same setting as MVSS-Net for fair comparison.
> > >
> > >   - **SAM pre-training uses additional data:** Our IMDPrompter leverages pre-trained SAM, which has already been trained on the large-scale segmentation dataset SA-1B. Therefore, while our approach explicitly uses CASIAv2, SAM's pre-training inherently includes additional data.
> > >
> > >   - **Performance with GlideCoco training:** As presented in Supplementary Table 18, we trained IMDPrompter on GlideCoco, achieving state-of-the-art (SOTA) performance on the corresponding test set.
> > >
> > >   - **Performance Alignment with UnionFormer's Training and Test Sets:** As shown in the table below, we used the same combined training set as UnionFormer and evaluated performance on each test set. Our method achieved state-of-the-art (SOTA) performance.
> > >
> > >   **Pixel-level F1 analysis:**
> > >
> > > |             |          |          | Optimal threshold |       |           |       |          |          | Fixed threshold(0.5) |       |           |       |
> > > |:-----------:|:--------:|:--------:|:-----------------:|:-----:|:---------:|:-----:|:--------:|:--------:|:--------------------:|:-----:|:---------:|:-----:|
> > > |             | Columbia | Coverage |       CASIA       |  NIST | CoCoGlide |  AVG  | Columbia | Coverage |         CASIA        |  NIST | CoCoGlide |  AVG  |
> > > |  CAT-Net v2 |   92.3   |   58.2   |       85.2        | 41.7  |   60.3    | 67.5  |   85.9   |   38.1   |         75.2         | 30.8  |   43.4    | 54.7  |
> > > |    Trufor   |   91.4   |   73.5   |       82.2        | 47.0  |   72.0    | 73.2  |   85.9   |   60.0   |         73.7         | 39.9  |   52.3    | 62.4  |
> > > | UnionFormer |   92.5   |   72.0   |       86.3        | 48.9  |   74.2    | 74.8  |   86.1   |   59.2   |         76.0         | 41.3  |   53.6    | 63.2  |
> > > | IMDPrompter |   92.7   |   74.0   |       87.1        | 50.1  |   74.0    | 75.6  |   86.7   |   60.3   |         76.8         | 42.4  |   52.9    | 63.8  |
> > >
> > >   **Image-level detection analysis:**
> > >
> > > |             |          |          | Image-level AUC |        |           |        |          |          | Accuracy |        |           |        |
> > > |:-----------:|:--------:|:--------:|:---------------:|:------:|:---------:|:------:|:--------:|:--------:|:--------:|:------:|:---------:|:------:|
> > > |             | Columbia | Coverage |      CASIA      |  NIST  | CoCoGlide |   AVG  | Columbia | Coverage |   CASIA  |  NIST  | CoCoGlide |   AVG  |
> > > |  CAT-Net v2 |  0.977   |  0.680   |      0.942      | 0.750  |   0.667   | 0.803  |  0.803   |  0.635   |  0.838   | 0.597  |   0.580   | 0.691  |
> > > |    Trufor   |  0.996   |  0.770   |      0.916      | 0.760  |   0.752   | 0.839  |  0.984   |  0.680   |  0.813   | 0.662  |   0.639   | 0.756  |
> > > | UnionFormer |  0.998   |  0.783   |      0.951      | 0.793  |   0.797   | 0.864  |  0.979   |  0.694   |  0.843   | 0.680  |   0.682   | 0.776  |
> > > | IMDPrompter |  0.998   |  0.791   |      0.962      | 0.801  |   0.801   | 0.871  |  0.984   |  0.703   |  0.851   | 0.682  |   0.680   | 0.780  |
> > >
> > >   **Pixel-level AUC analysis:**
> > >
> > > |             | Columbia | Coverage |  CASIA |  NIST  |   IMD  |   AVG  |
> > > |:-----------:|:--------:|:--------:|:------:|:------:|:------:|:------:|
> > > |    Trufor   |  0.947   |  0.925   | 0.957  | 0.877  |    -   | 0.927  |
> > > | UnionFormer |  0.989   |  0.945   | 0.972  | 0.881  | 0.860  | 0.929  |
> > > | IMDPrompter |  0.990   |  0.948   | 0.978  | 0.890  | 0.864  | 0.934  |
> > > ---

---

> ### Author Response · Authors · 2024-11-23
>
> **Weakness 6：Table 2 presents numerous NaN values for Sensitivity, Specificity, and F1 scores related to TruFor. Can the authors provide the corresponding metrics?**
>
> - Thank you for your constructive suggestions. We have supplemented the missing data in Table 2 of the main text, as shown below:
>
> |             |  |    CASIA   |       |       |   |    COVER   |       |       |  |  Columbia    |      |      |    |   IMD    |       |       |   MEAN  |       |
> |:-----------:|:-----:|:-----:|:-----:|:-----:|:------:|:-----:|:-----:|:-----:|:--------:|:----:|:----:|:----:|:-----:|:-----:|:-----:|:-----:|:-------:|:-----:|
> |             |  AUC  |  Sen. |  Spe. |   F1  |   AUC  |  Sen. |  Spe. |   F1  |    AUC   | Sen. | Spe. |  F1  |  AUC  |  Sen. |  Spe. |   F1  |   AUC   |   F1  |
> |    Trufor   | 0.916 | 72.6  | 81.3  | 75.1  | 0.770  | 96.1  | 43.9  | 61.3  |  0.996   | 97.6 | 92.3 | 93.6 | 0.669 | 86.7  | 70.6  | 58.6  | 0.8378  | 72.15 |
> | IMDPrompter | 0.978 | 91.6  | 99.3  | 77.3  | 0.796  | 96.8  | 61.3  | 70.3  |  0.983   | 97.3 | 91.6 | 93.6 | 0.671 | 71.6  | 73.2  | 63.7  | 0.8570  | 76.23 |
>
>
> **Weakness 7：Can you provide more recent methods for comparison, such as UnionFormer [3]?**
>
> - Thank you for your constructive suggestions.Referring to the response to Weakness 5, we have supplemented the comparison with UnionFormer, as detailed above.

---

> ### Comment · Reviewer_z9xA · 2024-11-25
>
> The first two tables, which present Pixel-level F1 analysis and Image-level detection analysis, include metrics such as Columbia, Coverage, CASIA, and NIST, that are higher than those reported in the submitted manuscript. However, the CoCoGlide metric is consistent with the value shown in Table 18 of the supplementary material. I am uncertain whether the authors re-trained the model and achieved improved performance. The above tables can not convince me.

---

> ### Author Response · Authors · 2024-11-26
>
> First, we must clarify that for the data added during the Rebuttal process to align with the experimental settings of UnionFormer [1], we have retrained our model strictly following UnionFormer's experimental configurations for the training and validation sets.
>
> We also appreciate your courageous attitude in questioning our work. Below is a detailed explanation addressing your concerns:
>
> **For the metrics on datasets such as Columbia, Coverage, CASIA, and NIST, the reason why the data supplemented in the Rebuttal are higher than those in Table 1 and Table 2 of the original paper is as follows:** Referring to UnionFormer, the training set includes five parts:
>
> 1. CASIA v2,
> 2. Fantastic Reality,
> 3. Tampered COCO, derived from COCO 2017 datasets,
> 4. Tampered RAISE, constructed based on the RAISE dataset, and
> 5. Pristine images selected from the COCO 2017 and RAISE datasets.
>
> This is the training set we used in the supplementary experiments during the Rebuttal process. However, in Table 1 and Table 2 of the original paper, we only used CASIA v2 as our training set (to align with the training set settings of MVSS-Net [2]; our data in Table 1 and Table 2 are consistent with MVSS-Net). Since Columbia, Coverage, CASIA, and NIST are all based on non-generative image manipulation methods, **methods like Trufor and CAT-Netv2 can benefit from training sets constructed with more non-generative image manipulation data.** Therefore, the metrics on these datasets supplemented in our Rebuttal are higher.
>
> **Regarding the CoCoGlide dataset, the reason why the data for methods like Trufor supplemented in the Rebuttal are similar to those in Supplementary Material Table 18 is as follows:** According to the UnionFormer paper, the performance of methods like Trufor on the CoCoGlide dataset did not improve even when using a larger training set (a larger dataset composed of five parts including CASIA v2, Fantastic Reality, and Tampered RAISE). This is because CoCoGlide is a dataset constructed based on generative image manipulation methods (there is a significant domain shift compared to training sets constructed with non-generative methods), **so methods like CAT-Netv2 and Trufor cannot benefit from additional non-generative image manipulation training data.** However, our IMDPrompter behaves differently. On one hand, using more non-generative image manipulation training data, our IMDPrompter shows a slight decrease in pixel-level metrics such as P-F1. At the same time, it shows a slight improvement in image-level metrics such as I-AUC. This indicates that IMDPrompter has different learning behaviors compared to methods like Trufor when utilizing more non-generative image manipulation training data.
>
> [1] Li, S., Ma, W., Guo, J., Xu, S., Li, B., & Zhang, X. (2024). UnionFormer: Unified-Learning Transformer with Multi-View Representation for Image Manipulation Detection and Localization. In *Proceedings of the IEEE/CVF Conference on Computer Vision and Pattern Recognition* (pp. 12523-12533).
>
> [2] Chen, X., Dong, C., Ji, J., Cao, J., & Li, X. (2021). Image manipulation detection by multi-view multi-scale supervision. In *Proceedings of the IEEE/CVF International Conference on Computer Vision* (pp. 14185-14193).

---

> > ### Author Response · Authors · 2024-12-01
> > **Looking forward to your reply (less than 48 hours)**
> >
> > Dear reviewer z9xA,
> >
> > We hope our responses have adequately addressed your previous concerns. The discussion period is approaching the end in 48 hours. If you have any additional comments or questions, please feel free to share them. Your feedback is greatly appreciated.
> >
> > Sincerely,
> >
> > Authors

---

> > ### Comment · Reviewer_z9xA · 2024-12-02
> >
> > Thank you for your response and the detailed rebuttals.
> >
> > In the supplementary material of the original paper, the CoCoGlide metric is based on your proposed model trained on the CASIA v2 dataset. However, after one month (the rebuttal period), the authors re-trained the same model with additional datasets (five datasets), and the exact same CoCoGlide values were reported. This raises doubts about whether the authors actually re-trained the same model on more datasets.
> >
> > Would the authors be willing to provide the code and pre-trained models for both the original paper version and the rebuttal version so that we can validate the reported metrics?

---

> > > ### Author Response · Authors · 2024-12-04
> > >
> > > ### Dear Reviewer z9xA,
> > >
> > > Thank you for your valuable feedback. Regarding the experimental results on the CoCoGlide dataset, the following explanation is provided:
> > >
> > > Referring to the original data in **Trufor (CVPR 2023)** [1], the performance of **Trufor** and **CATNetv2** trained on CASIA v2, evaluated on the CoCoGlide dataset, is as follows:
> > >
> > > | Methods      | CoCoGlide | P-F1 (best) | P-F1 (fixed) | I-AUC | I-Acc |
> > > |--------------|-----------|-------------|--------------|-------|-------|
> > > | CAT-Net v2   | 0.603     | 0.434       | 0.667        | 0.580 |       |
> > > | Trufor       | 0.720     | 0.523       | 0.752        | 0.639 |       |
> > >
> > > The **IMDPrompter**'s supplementary material (Table 18) uses the same experimental settings as Trufor and achieves similar performance.
> > >
> > > Referring to the original data in **UnionFormer (CVPR 2024)** [2], the performance of **Trufor**, **CATNetv2**, and **UnionFormer** trained on CASIA v2 and five additional datasets, evaluated on the CoCoGlide dataset, is as follows:
> > >
> > > | Methods      | CoCoGlide | P-F1 (best) | P-F1 (fixed) | I-AUC | I-Acc |
> > > |--------------|-----------|-------------|--------------|-------|-------|
> > > | CAT-Net v2   | 0.603     | 0.434       | 0.667        | 0.580 |       |
> > > | Trufor       | 0.720     | 0.523       | 0.752        | 0.639 |       |
> > > | UnionFormer  | 0.742     | 0.536       | 0.797        | 0.682 |       |
> > >
> > > The experiments conducted during the rebuttal process, using the same experimental settings as UnionFormer, show similar performance.
> > >
> > > In summary, the experimental results reported in both the original manuscript and the rebuttal regarding CoCoGlide are aligned with the results from **Trufor (CVPR 2023)** and **UnionFormer (CVPR 2024)** under the same experimental settings.
> > >
> > > We greatly appreciate your attention and recognition of our research. Your concerns about the experimental details are fully understood. Both the manuscript and rebuttal thoroughly describe the experimental design, analysis, results, as well as the data analysis and processing process. If there are any specific questions regarding the data, further clarification will be provided. Additionally, we commit to making all code publicly available after the paper is accepted.
> > >
> > > #### References
> > > [1] Guillaro, F., Cozzolino, D., Sud, A., Dufour, N., & Verdoliva, L. (2023). Trufor: Leveraging all-round clues for trustworthy image forgery detection and localization. In *Proceedings of the IEEE/CVF conference on computer vision and pattern recognition* (pp. 20606-20615).
> > >
> > > [2] Li, S., Ma, W., Guo, J., Xu, S., Li, B., & Zhang, X. (2024). UnionFormer: Unified-Learning Transformer with Multi-View Representation for Image Manipulation Detection and Localization. In *Proceedings of the IEEE/CVF Conference on Computer Vision and Pattern Recognition* (pp. 12523-12533).

---

### Author Response · Authors · 2024-12-04
**The General Response**

We would like to express our sincere gratitude for the consistent positive feedback and valuable suggestions from all the reviewers. First, we appreciate the reviewers' recognition of our work in the following areas:

- **Innovative Application:**
The paper applies SAM to the underexplored area of image manipulation detection, which extends SAM's use case beyond traditional segmentation tasks. The proposal of a cross-view automated prompt learning paradigm with IMDPrompter is unique and addresses the challenges specific to manipulation detection tasks. [t8Py][tfmM][tXob]

- **Automation of Prompt Generation:**
One of the standout innovations is the elimination of SAM's reliance on manual prompts. The introduction of modules like Optimal Prompt Selection (OPS) and Cross-View Prompt Consistency (CPC) strengthens SAM’s utility by automating the prompt generation, potentially making SAM more accessible for manipulation detection. [t8Py][tfmM][tXob]

- **Robustness and Generalizability:**
IMDPrompter's multi-view approach—integrating RGB, SRM, Bayer, and Noiseprint—demonstrates enhanced generalization, particularly on out-of-domain datasets. The ablation studies further substantiate the contributions of each module, which supports the validity of the multi-view and prompt-learning design. [tfmM][tXob]

- **Strong Experimental Validation:**
The model shows significant improvements in image-level and pixel-level metrics across multiple datasets (CASIA, Columbia, IMD2020, etc.), indicating its robustness. The experimental setup includes various metrics (AUC, F1-scores), highlighting the model’s strengths compared to prior approaches. [t8Py][tfmM][tXob]

## Response to Reviewers' Suggestions

In response to the reviewers' feedback, we conducted the following discussions:

- **Complexity and Computational Costs:**
  Following the suggestions from reviewers [t8Py][tfmM], we have provided a more detailed discussion on the complexity and computational costs.

- **Ablation Studies on View Impact:**
  Following the suggestions from reviewers [z9xA][t8Py][tfmM][tXob], we have expanded the discussion on ablation studies related to the impact of each view.

- **Comparison with UnionFormer:**
  Following the suggestion from reviewers [z9xA][tXob], we have included a discussion comparing our approach with **UnionFormer**.

- **Application to Other Modalities (e.g., Video Manipulation Detection):**
  Following the suggestion from reviewers [t8Py][tfmM], we have discussed the specific challenges and modifications required to apply IMDPrompter to other modalities, such as video manipulation detection.

We have carefully considered the feedback from each reviewer and will continue to improve the quality of our work.

---

### Meta-Review · Area_Chair_tb8P · 2024-12-23

**Metareview:**

The paper proposes IMDPrompter, an automated prompt learning framework based on the Segment Anything Model (SAM) for image manipulation detection (IMD). By leveraging multiple views, including RGB, SRM, Bayar, and Noiseprint, it integrates auxiliary masks and bounding boxes for SAM. Key modules, such as Cross-view Feature Perception and Prompt Mixing modules, enhance feature integration within the Mask Decoder. Evaluated on five datasets, IMDPrompter demonstrates performance improvements over previous methods in both in-distribution and out-of-distribution settings, showcasing its robustness and effectiveness in extending SAM's capabilities for IMD tasks.

Strength: The paper's strength lies in its interesting application of the SAM to the image manipulation detection task through a cross-view automated prompt learning paradigm. By integrating multiple views such as RGB, SRM, Bayar, and Noiseprint, the method enriches SAM’s utility with auxiliary masks and bounding boxes. Modules like Optimal Prompt Selection (OPS) and Cross-View Prompt Consistency (CPC) automate prompt generation, eliminating manual reliance and enhancing generalization across datasets. The straightforward design effectively combines semantic-agnostic features and delivers strong experimental validation with improved performance in image-level and pixel-level metrics.

Weakness: Reviewers raise the questions about the lack of clarity regarding the necessity of using SAM for image manipulation detection, the reliance on the accuracy of generated masks for prompts, and some missing technical details. Besides, claims regarding the effectiveness of OPS and CPC in enhancing alignment across views are insufficiently supported with evidence. Additionally, the method introduces increased complexity by incorporating multiple modules, encoders, and views without justifying the choice of specific views like SRM, Bayar, and Noiseprint or detailing how they contribute to mask precision. During the rebuttal, the author has provided an exhaustive response with additional ablation study and new results to address these concerns raised by the reviewers.

**Additional Comments On Reviewer Discussion:**

During the rebuttal, the author provided a very detailed response to all reviewers’ comments, which has well addressed most of the concerns. All reviewers have responded to the rebuttal and three reviewers have raised the score to reflect the paper improvement from the author’s response including additional ablation study and new results. After the rebuttal, all the scores have converged to around the borderline, while one reviewer still claims a negative trend due to the concern about the authenticity of the experiment results. During the rebuttal, one reviewer is asking if the authors are willing to provide the code and pre-trained models for both the original paper version and the rebuttal version so that the reviewer can validate the reported metrics. Although the code submission is not required for review and that the author commits to making all code publicly available after the paper is accepted is well acceptable, directly validating the code by reviewers may be stronger evidence. The decision is made after the discussion with the SAC, particularly considering how the exhaustive rebuttal has effectively addressed most of questions and concerns, which is also reflected by the score changes from three reviewers.

---

### Decision · Program_Chairs · 2025-01-22

Accept (Poster)